# Variational Inference for Uncertain Optimal Transport via Sinkhorn Parametrization

Ananyapam De [1]   Linus Bleistein [2]   Anton Frederik Thielmann [3] [*]   Benjamin Säfken [1]

## Abstract

Optimal Transport (OT) traditionally relies on a fixed ground cost to produce a single deterministic transport plan, a practice that overlooks the inherent variability and noise in real-world data. While recent sampling-based approaches to OT offer a principled way to quantify this uncertainty, they are computationally prohibitive and struggle to scale. In this paper, we introduce Sinkhorn-parameterized Variational Inference (SPVI), the first scalable variational framework for performing posterior inference over transport plans. Our key insight is that the Sinkhorn map can be treated as a differentiable reparameterization of the set of entropic plans. This enables the use of flexible generative models like normalizing flows to approximate distributions over transport plans while enforcing marginal constraints. We experimentally demonstrate that our method matches the quality of intensive sampling techniques at a fraction of the computational cost, scaling effectively to large-scale problems.

## 1. Introduction

Optimal transport (OT) is a powerful framework for comparing and transforming probability distributions, and has gained significant traction in machine learning for defining distributional distances and generative modeling metrics (Cuturi, 2013; Peyré & Cuturi, 2019). Classical OT assumes a fixed, known cost function (e.g. a ground metric between samples) and yields a single deterministic matching between points, or *transport plan*, that minimizes the total transportation cost. In many real-world scenarios, however, the cost

is not exact or deterministic and may be uncertain or randomly fluctuating. This is, for instance, the case in logistics (Agueh & Carlier, 2011), where transport costs might be affected by random traffic conditions; in single-cell biology (Schiebinger et al., 2019), where batch effects can perturb the computed costs; or when data are missing in tabular datasets (Bleistein et al., 2025).

When multiple realizations of the random matching process are available, taking advantage of this observed variability is crucial for downstream applications such as predictions made from the transport plan or assessing possible uncertainty about future matchings. Concretely, given novel samples to match, one might want to infer a *distribution* over possible transport plans rather than a single fixed plan. We therefore ask the following question:

*How can we take advantage of multiple observed noisy transport plans to learn a distribution over future matchings?*

As illustrated in Figure 1, traditional sampling methods can capture these distributions, but they are often computationally prohibitive for real-time applications. We tackle this problem by building upon advances in variational inference and deep generative modeling. Variational Inference (VI) (Hoffman et al., 2013; Ranganath et al., 2014) is a widely used approach to approximating complex posterior distributions by optimizing over a family of tractable distributions. A cornerstone of modern VI with continuous latent-variable models is the *reparameterization trick* (Kingma & Welling, 2022; Rezende et al., 2014), which enables low-variance gradient estimates by expressing random samples as deterministic functions of parameters and noise (as done in Variational Auto Encoders (VAE) (Kingma & Welling, 2019) and many Bayesian deep learning methods (Blundell et al., 2015; Gal & Ghahramani, 2016)). We bring this idea to OT by deriving an efficient reparameterization trick for the optimal transport problem, which in turn allows us to sample from a parametrized distribution over random plans.

In order to learn a flexible distribution over transport plans, we use normalizing flows. Normalizing flows (Rezende & Mohamed, 2015; Dinh et al., 2017) construct complex distributions by composing invertible transformations of a base

---

[*]Work done while at TU Clausthal. [1]Technische Universität Clausthal, Germany [2]Schools of Computer and Communication, and Life Sciences, EPFL, Lausanne, Switzerland [3]Amazon Music, Germany. Correspondence to: Ananyapam De <ananyapam.de@tu-clausthal.de>.

*Proceedings of the 43rd International Conference on Machine Learning*, Seoul, South Korea. PMLR 306, 2026. Copyright 2026 by the author(s).

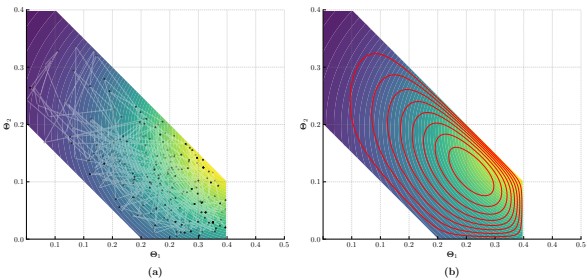

**Figure 1.** (a) 500 posterior samples on a $2 \times 3$ transport polytope generated via Hamiltonian Monte Carlo (HMC) with BayesOT. HMC suffers from high latency and poor mixing due to rigid marginal constraints. (b) SPVI (red contours) is fast and approximates the posterior with high fidelity, enabling efficient uncertainty quantification for optimal transport.

noise distribution. In principle, one could directly attempt to design a normalizing flow on the manifold of transport plans using a parameterization that guarantees the Jacobian preserves the polytope constraints. However, defining flows that exactly respect marginal constraints is highly nontrivial. Instead, our approach uses the Sinkhorn mapping (Knight, 2007), a non-invertible but structured iterative scaling procedure that transforms a strictly positive matrix into a coupling with specified marginals by alternating between row and column normalization until convergence. The unconstrained Sinkhorn map is non-injective because diagonal rescalings of the input kernel leave the resulting coupling unchanged. We remove this by working in a gauge-fixed log-kernel subspace, where the Sinkhorn map becomes a smooth global chart of the relative interior of the transport polytope, allowing us to combine flexible normalizing flows with exact marginal constraint satisfaction. The resulting variational family has a tractable density and can therefore be optimized using the ELBO (Huszár, 2017; Uppal et al., 2023).

**Contributions.** Our contributions are as follows.

1. We introduce **Sinkhorn-parameterized Variational Inference** (SPVI), a variational framework for posterior inference over the transport polytope.

2. We prove that the truncated Sinkhorn operator and its associated Jacobian converge geometrically to their exact counterparts, justifying the use of finite-step iterations in the ELBO.

3. We empirically show the scalability of SPVI compared to sampling-based approaches.

4. We demonstrate the utility of our approach in tasks such as domain adaptation, joint representation and transport learning, and single-cell data analysis.

## 2. Related Work

**Optimal Transport and Entropic Regularization.** OT has become a cornerstone of modern machine learning and has been embraced as a means to compare distributions and define generative model objectives (Cuturi, 2013; Arjovsky et al., 2017; Peyré & Cuturi, 2019). Classical OT seeks a coupling $\Gamma$ that minimizes the expected cost $\langle C, \Gamma \rangle$ on the transport polytope $\Pi(\mu, \nu)$ (defined in Sec. 3). Without additional regularization, the solution to the OT problem is often sparse and hard to compute at scale. Variants of *regularized* OT have since been proposed to alleviate this computational burden and enforce desirable structural properties, including quadratic, group-sparse, and Tsallis regularizations. We focus in particular on *entropy-regularized* OT, which adds a convex entropy penalty scaled by a regularization parameter $\varepsilon > 0$. This relaxation yields a smoother, unique optimal plan that can be solved efficiently via the Sinkhorn–Knopp matrix scaling algorithm. Entropic OT not only accelerates OT computations by orders of magnitude but also admits a probabilistic interpretation: it is equivalent to an entropy-minimization (Schrödinger bridge) problem (Léonard, 2014; Bortoli et al., 2021; Bunne et al., 2023a). Specifically, the optimal coupling is the KL projection of a Gibbs kernel $K_\varepsilon = \exp(-C/\varepsilon)$ onto $\Pi(\mu, \nu)$, representing the most likely coupling subject to marginal constraints (Cuturi, 2013; Léonard, 2014; Stromme, 2023).

While regularization provides a smoothed objective, it does *not* provide a full Bayesian treatment of uncertainty and the resulting solution is a point estimate that fails to quantify sensitivity to varying costs. Our work interprets the regularizer as a negative log-prior, transforming the transport problem from a single optimization task into a posterior inference task over the entire polytope.

**Bayesian Optimal Transport and Uncertain Costs.** Uncertain OT has been addressed by a few recent works. The BayesOT framework of Mallasto et al. (2021) is the first to explicitly infer a posterior over transport plans given a stochastic cost function. They define a prior distribution on the space of couplings and use a likelihood based on the cost observations, resulting in a posterior that can be sampled via Markov chain Monte Carlo. Their work demonstrated that one can obtain meaningful uncertainty quantification for OT, for example by producing credible intervals for entries of the transport matrix in a demographic matching problem, and that the maximum a posteriori (MAP) coincides with entropy-regularized solutions under appropriate priors. Nevertheless, a limitation of BayesOT is the computational overhead and the long mixing times of MCMC. In fact, earlier attempts at Bayesian inference on contingency tables (joint distributions) with fixed marginals, such as the work of Rosen et al. (2001), encounter convergence issues when using naive Gibbs samplers, precisely because of the rigid

marginal constraints. Frogner & Poggio (2019) addresses a related problem by proposing a family of models with a prior over joint distributions and deriving an *exact* MAP solution via Dijkstra's algorithm. Their approach yields a point estimate of the coupling that maximizes the posterior, thus generalizing OT to incorporate prior knowledge without sacrificing the marginal constraints.

In essence, we blend the efficiency of the Sinkhorn updates and projections (as used by Frogner & Poggio for MAP inference) with a variational approximation that captures the posterior uncertainty as BayesOT intended. Recent studies in distributionally robust optimization (De et al., 2024; Blanchet et al., 2021; Mohajerin Esfahani & Kuhn, 2018; Mukherjee et al., 2021) and robust OT (Balaji et al., 2020) have also considered worst-case or adversarial cost perturbations (Sinha et al., 2018), but these typically yield a single robust coupling rather than a distribution of couplings.

**Differentiable Optimal Transport.** Our work crucially takes advantage of the differentiability of the Sinkhorn algorithm. This was also exploited by Gumbel–Sinkhorn networks (Mena et al., 2018), which approximate permutation matrices in a differentiable manner by adding Gumbel noise to a cost matrix and applying Sinkhorn normalization. In our case, we do not restrict to permutations, as our aim is to sample general couplings, but we similarly rely on the Sinkhorn algorithm's ability to produce doubly stochastic matrices as outputs of a neural network layer and on its differentiability. Recent research has further incorporated Sinkhorn layers in neural architectures, for example as differentiable relaxations for ranking, sorting, and related permutation-based operators (Adams & Zemel, 2011; Cuturi et al., 2019; Tay et al., 2020), to learn flexible weight-sharing patterns via learnable doubly-stochastic matrices (van der Linden et al., 2024), or to enforce structural constraints by constraining learned mixing matrices to lie in the Birkhoff polytope via Sinkhorn–Knopp normalization (Yang & Gao, 2026).

## 3. Background on Optimal Transport

We recall the basic notions of OT and its regularized variants. We work throughout in the finite, discrete setting.

**Discrete optimal transport.** Let $\mu \in \Delta^{n-1}_{++}$ and $\nu \in \Delta^{m-1}_{++}$ be probability vectors on finite sets $\{x_1, \ldots, x_n\}$ and $\{y_1, \ldots, y_m\}$, so that $\mu_i > 0$, $\sum_{i=1}^{n} \mu_i = 1$ and similarly for $\nu$. We consider *cost functions*

$$c : \{x_1, \ldots, x_n\} \times \{y_1, \ldots, y_m\} \to \mathbb{R}_+$$

which give rise to nonnegative cost matrices $C \in \mathbb{R}^{n \times m}_+$ with entries $C_{ij} = c(x_i, y_j)$. A *transport plan* (or *coupling*) between $\mu$ and $\nu$ is a nonnegative matrix $\Gamma \in \mathbb{R}^{n \times m}_+$ whose row and column sums coincide with $\mu$ and $\nu$:

$$\Pi(\mu, \nu) := \left\{ \Gamma \in \mathbb{R}^{n \times m}_+ : \Gamma \mathbf{1}_m = \mu, \ \Gamma^\top \mathbf{1}_n = \nu \right\}, \quad (1)$$

where $\mathbf{1}_k$ denotes the $k$-dimensional vector of ones. The quantity $\Gamma_{ij}$ represents the amount of mass moved from $x_i$ to $y_j$. The set $\Pi(\mu, \nu)$ is a convex polytope in $\mathbb{R}^{n \times m}$, usually referred to as the *transport polytope*.

Given a cost matrix $C$, the *Kantorovich optimal transport problem* seeks a coupling of minimal total cost:

$$\mathrm{OT}(\mu, \nu; C) := \min_{\Gamma \in \Pi(\mu, \nu)} \langle C, \Gamma \rangle \quad (2)$$

where $\langle A, B \rangle = \sum_{ij} A_{ij} B_{ij}$ is the Frobenius inner product. Any minimizer $\Gamma^\star \in \Pi(\mu, \nu)$ is called an *optimal transport plan*. When $c$ arises from a metric $C_{ij} = d(x_i, y_j)^p$, the corresponding continuum analogue satisfies $\mathrm{OT}(\mu, \nu; d^p) = \mathcal{W}_p(\mu, \nu)^p$, where $\mathcal{W}_p$ is the $p$-Wasserstein distance. In this paper we remain in the discrete setting and focus on the structure of $\Pi(\mu, \nu)$ and on the effect of cost uncertainty.

**Geometry of the transport polytope.** The transport polytope $\Pi(\mu, \nu)$ is the intersection of the non-negative orthant $\mathbb{R}^{n \times m}_+$ with an affine subspace determined by the marginal constraints. Let $\mathcal{A}$ denote this affine subspace of matrices with fixed row and column sums:

$$\mathcal{A} := \left\{ \Gamma \in \mathbb{R}^{n \times m} : \Gamma \mathbf{1}_m = \mu, \ \Gamma^\top \mathbf{1}_n = \nu \right\}.$$

Thus, we can compactly write $\Pi(\mu, \nu) = \mathcal{A} \cap \mathbb{R}^{n \times m}_+$.

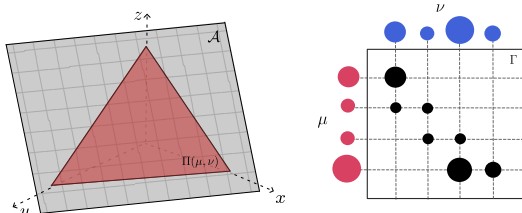

**Figure 2.** Geometric and matrix representations of the transport polytope. (Left) The transport polytope $\Pi(\mu, \nu)$ (red) is the intersection of the non-negative orthant with the affine subspace $\mathcal{A}$ defined by the marginal constraints. (Right) A discrete transport plan $\Gamma$ visualized as a coupling matrix; the sizes of the black circles represent the mass values $\Gamma_{ij}$, while the marginals $\mu$ (red) and $\nu$ (blue) are obtained by summing the rows and columns, respectively.

While $\mathcal{A}$ is defined by $n + m$ linear equalities, the constraints are not independent due to the mass conservation $\sum_i \mu_i = \sum_j \nu_j = 1$. One constraint is redundant, making the codimension of $\mathcal{A}$ equal to $n + m - 1$. Consequently, $\dim(\mathcal{A}) := r = (n-1)(m-1)$. Since we assume strictly positive marginals, $\Pi(\mu, \nu)$ possesses a relative interior

$$\mathcal{M} := \left\{ \Gamma \in \Pi(\mu, \nu) : \Gamma_{ij} > 0 \ \forall i, j \right\} \quad (3)$$

which is an open subset of $\mathcal{A}$ with $\dim(\mathcal{M}) = r$. As the Kantorovich problem (2) is a linear program, its solutions are guaranteed to exist and typically lie on this boundary,

yielding plans with at most $n + m - 1$ non-zero entries. Crucially, the vertices and facets of the boundary $\partial\Pi(\mu, \nu)$ represent sparse transport plans.

**Regularized optimal transport.** While mathematically elegant, this sparsity makes the solution highly sensitive to perturbations in the cost matrix $C$, typically non-differentiable with respect to parametrizations of the cost matrices, and computationally expensive to obtain for high-dimensional distributions. This motivates *regularized* OT problems of the form

$$\mathrm{OT}_R(\mu, \nu; C) := \min_{\Gamma \in \Pi(\mu,\nu)} \left\{ \langle C, \Gamma \rangle + R(\Gamma) \right\}, \quad (4)$$

where $R : \mathbb{R}^{n \times m} \to \mathbb{R} \cup \{+\infty\}$ is a strictly convex regularizer. Under strict convexity of $R$ on $\Pi(\mu, \nu)$, problem (4) admits a unique minimizer (Peyré & Cuturi, 2019), which we denote by

$$\Gamma_R^\star(\mu, \nu; C) \in \Pi(\mu, \nu).$$

A particularly important choice is the (negative) entropy regularizer

$$H(\Gamma) := -\sum_{i,j} \Gamma_{ij} \log \Gamma_{ij}, \qquad R(\Gamma) = -\varepsilon\, H(\Gamma), \quad (5)$$

with $\varepsilon > 0$. The corresponding problem

$$\mathrm{OT}_\varepsilon(\mu, \nu; C) := \min_{\Gamma \in \Pi(\mu,\nu)} \left\{ \langle C, \Gamma \rangle - \varepsilon H(\Gamma) \right\} \quad (6)$$

is the now standard *entropic* optimal transport problem. Since the gradient of $H(\Gamma)$ diverges as any $\Gamma_{ij} \to 0$, the entropy pulls the solution away from the sparse boundary $\partial\Pi(\mu, \nu)$ and into the relative interior $\mathcal{M}$. Its unique solution $\Gamma_\varepsilon^\star$ can be computed using the classical Sinkhorn–Knopp algorithm (see Section 4) by iterative row/column scaling.

**Gibbs posteriors over couplings.** Suppose we observe a collection of cost matrices $\mathcal{C} = \{C^{(1)}, \dots, C^{(N)}\} \subset \mathbb{R}_+^{n \times m}$ and wish to quantify the uncertainty this induces in the transport plan. A natural object of interest is then a *distribution* over couplings $\Gamma \in \Pi(\mu, \nu)$ that combines prior structural preferences with information from the observed costs. A convenient way to encode such a distribution is via a *Gibbs measure* on $\Pi(\mu, \nu)$ with an energy functional $E(\Gamma)$. All densities are defined with respect to the $r$-dimensional Hausdorff measure $\mathcal{H}^r$ on the affine hull of the polytope. When $R(\Gamma) = -\log p(\Gamma \mid \mu, \nu)$, these measures correspond to Bayesian posteriors. This leads to two natural energy functionals corresponding to different posterior behaviors, as employed by Mallasto et al. (2021). The $L_1$ optimality notion uses the energy functional

$$E(\Gamma) := R(\Gamma) + \beta \sum_{k=1}^{N} \langle C^{(k)}, \Gamma \rangle,$$

where $\beta > 0$ is an inverse temperature parameter. The resulting density is

$$p(\Gamma \mid C^{(1:N)}, \mu, \nu) \propto \exp(-E(\Gamma))\, \mathbf{1}_{\Pi(\mu,\nu)}$$

The MAP estimator for this posterior corresponds to the standard regularized OT solution:

$$\Gamma_{\mathrm{MAP}} = \arg\min_{\Gamma \in \Pi(\mu,\nu)} \{ R(\Gamma) + \beta \langle \bar{C}, \Gamma \rangle \}$$

Because this depends only on the aggregate cost $\bar{C}$, it cannot capture multimodality arising from conflicting cost realizations. To capture regime-dependent structures, the $L_2$ construction uses a log-sum-exp aggregation:

$$E(\Gamma) := R(\Gamma) - \log \left( \sum_{k=1}^{N} \exp(-\beta \langle C^{(k)}, \Gamma \rangle) \right)$$

The associated density simplifies to a mixture-like form:

$$p(\Gamma \mid C^{(1:N)}) \propto \exp(-R(\Gamma)) \sum_{k=1}^{N} \exp(-\beta \langle C^{(k)}, \Gamma \rangle) \mathbf{1}_{\Pi(\mu,\nu)} \tag{7}$$

This allows the posterior to capture multimodality when different cost matrices favor distinct transport plans. BayesOT defines a likelihood via auxiliary binary optimality variables and performs posterior inference by HMC/NUTS sampling on a constrained chart (Brubaker et al., 2012) of the transport polytope to respect the marginal constraints, which becomes costly at scale (Beskos et al., 2011; Brooks et al., 2011), with multimodality further increasing autocorrelation (Woodard et al., 2009). Variational inference replaces this repeated constrained sampling with differentiable optimization of a reparameterized family.

## 4. Sinkhorn–parameterized variational inference

We approximate the Gibbs posterior (7) over couplings $\Gamma \in \Pi(\mu, \nu)$ by a reparameterized variational family supported on $\mathcal{M}$.

**Definition 4.1.** (Sinkhorn, 1964) *The Sinkhorn operator* $\mathcal{S} : \mathbb{R}_{++}^{n \times m} \times \Delta_{++}^{n-1} \times \Delta_{++}^{m-1} \to \mathcal{M}$ *maps a strictly positive kernel $K$ to the unique matrix $\Gamma = \mathrm{diag}(u) K \mathrm{diag}(v)$ in the transport polytope $\Pi(\mu, \nu)$. The scaling vectors $u \in \mathbb{R}_{++}^n$ and $v \in \mathbb{R}_{++}^m$ are defined as the unique (up to a scalar factor) fixed points of the alternating scaling steps:*

$$u \leftarrow \frac{\mu}{Kv} \quad and \quad v \leftarrow \frac{\nu}{K^\top u}, \tag{8}$$

*where the division is performed element-wise.*

To construct a valid density on $\mathcal{M}$, we must define a map that naturally enforces the marginal constraints while handling the over-parameterization of the logit space.

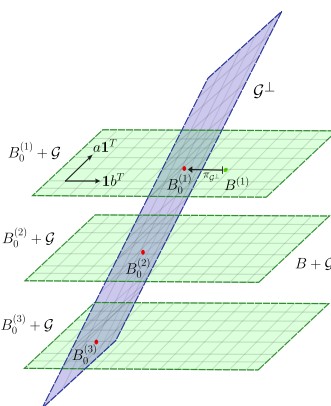

**Figure 3.** The Sinkhorn map is invariant to additive row/column offsets in log space: $B \mapsto B + a\mathbf{1}_m^\top + \mathbf{1}_n b^\top$, which generates a gauge orbit $B + \mathcal{G}$ shown as a family of parallel green planes. The blue plane denotes the gauge slice $\mathcal{G}^\perp$. Each orbit intersects $\mathcal{G}^\perp$ in exactly one point $B_0^{(k)}$, providing a unique gauge-fixed representative. The arrow is the projection $B^{(1)} \xrightarrow{\pi_{\mathcal{G}^\perp}} B_0^{(1)}$.

**Gauge-fixed logit embedding.** The Sinkhorn operator $\mathcal{S}(K; \mu, \nu)$ is invariant to diagonal rescaling of the kernel, meaning $\mathcal{S}(K) = \mathcal{S}(D_1 K D_2)$ for any diagonal matrices $D_1, D_2$ with positive entries. In the log-domain, where $B = \log K$, this invariance takes the form of additive gauge freedom: adding a row-wise or column-wise constant to $B$ does not change the resulting coupling. We define the subspace of these redundant additive perturbations as

$$\mathcal{G} := \{a\mathbf{1}_m^\top + \mathbf{1}_n b^\top : a \in \mathbb{R}^n, b \in \mathbb{R}^m\}.$$

The pre-image of any single coupling $\Gamma \in \mathcal{M}$ under the log-Sinkhorn map is an affine subspace of the form $B + \mathcal{G}$. As illustrated in Figure 3, to ensure bijectivity, we restrict our logits to the orthogonal complement $\mathcal{G}^\perp = \{B \in \mathbb{R}^{n \times m} : B\mathbf{1}_m = 0, B^\top \mathbf{1}_n = 0\}$. This choice is without loss of generality, as every equivalence class in the logit space modulo $\mathcal{G}$ has exactly one representative in $\mathcal{G}^\perp$.

**Proposition 4.2.** *Let $\mathcal{G} = \{a\mathbf{1}_m^\top + \mathbf{1}_n b^\top : a \in \mathbb{R}^n, b \in \mathbb{R}^m\}$ be the subspace of additive gauge freedoms, and let $\mathcal{G}^\perp$ be its orthogonal complement in $\mathbb{R}^{n \times m}$. Let $Q \in \mathbb{R}^{nm \times r}$ be a matrix with orthonormal columns spanning $\mathrm{vec}(\mathcal{G}^\perp)$. Define the map $\Psi : \mathbb{R}^r \to \mathcal{M}$ by*

$$\Psi(z) := \mathcal{S}\big(\exp(\mathrm{mat}(Qz)); \mu, \nu\big) \tag{9}$$

*where $\mathrm{mat}(\cdot)$ is the matricization operator that reshapes a vector in $\mathbb{R}^{nm}$ into a matrix $\mathbb{R}^{n \times m}$. Then $\Psi$ is a smooth global diffeomorphism from $\mathbb{R}^r$ onto $\mathcal{M}$. In particular, it is globally injective and its Jacobian $J_\Psi(z)$ has full column rank $r$ everywhere.*

*Proof.* See Appendix D. $\square$

**Sampler.** We define the reparameterized sampler $\Gamma$ via the following sequence of transformations. We first sample $\xi \sim p(\xi) = \mathcal{N}(0, I_r)$, and apply an invertible normalizing flow $f_\phi : \mathbb{R}^r \to \mathbb{R}^r$, conditioned on $\mathcal{C}$ with tractable Jacobian determinant. To ensure numerical stability, we apply a bounded squashing map

$$s(x) = c \tanh(x/c)$$

to obtain a latent code $z = h_\phi(\xi) = s(f_\phi(\xi)) \in (-c, c)^r$. We then map $z$ to the gauge-fixed logit space via $Q$ as

$$B(z) := \mathrm{mat}(Qz) \in \mathcal{G}^\perp.$$

Finally, we exponentiate to obtain the kernel $K = \exp(B(z)) \in \mathbb{R}_{++}^{n \times m}$ and apply the Sinkhorn scaling operator $\mathcal{S}(\cdot; \mu, \nu)$ to enforce marginals:

$$\Gamma(\xi) := \mathcal{S}\big(\exp\big(B(h_\phi(\xi))\big); \mu, \nu\big) \in \mathcal{M}. \tag{10}$$

In practice, $\mathcal{S}$ is implemented via a fixed number of iterations $T$ (see Appendix C) chosen so that the marginal violation is below a prescribed tolerance. We give a description of our sampling procedure in Algorithm 1.

### 4.1. Variational objective and density on $\mathcal{M}$

We optimize the variational parameters $\phi$ by maximizing the ELBO. Let $\tilde{F}(z) := \mathcal{S}\big(\exp(\mathrm{mat}(Qz)); \mu, \nu\big)$ denote the map from the squashed latent code to the coupling. The density of the pushforward distribution $q_\phi$ on the manifold $\mathcal{M}$ is obtained via the area formula.

Because $\tilde{F}$ is a globally injective immersion, the area formula applies without summation over preimages. The pushforward density $q_\phi$ with respect to $\mathcal{H}^r$ on $\mathcal{M}$ is:

$$\log q_\phi(\Gamma \mid \mathcal{C}) = \log p(\xi) - \log |\det J_{h_\phi}(\xi)| \\ - \frac{1}{2} \log \det (G(z))$$

The term $G(z) = J_{\tilde{F}}(z)^\top J_{\tilde{F}}(z)$ is the Gram matrix and the factor $\sqrt{\det G(z)}$ measures the local volume expansion induced by the embedding. Since $\tilde{F}$ is an immersion, $J_{\tilde{F}}$ has full rank $r$, ensuring the Gram matrix is positive definite.

Substituting this density into the standard ELBO definition, we obtain the tractable objective (up to an additive constant independent of $\phi$):

$$\mathcal{L}(\phi) \propto \mathbb{E}_{\xi \sim \mathcal{N}(0, I_r)}\Big[-E(\Gamma(\xi)) + \log |\det J_{h_\phi}(\xi)| \\ + \frac{1}{2} \log \det G(z)\Big]. \tag{11}$$

We use $S$ Monte Carlo samples of $\xi$ to approximate the expectation cheaply. Since $\log \det G(z) = \mathrm{Tr}(\log G(z))$, we can use the Hutchinson trace estimator (Hutchinson,

---

**Algorithm 1** Sinkhorn-parameterized Variational Inference

---

**Input:** marginals $\mu, \nu$; costs $\mathcal{C}$; energy functional $E(\cdot)$; flow $f_\phi$; gauge basis $Q$; Sinkhorn iterations $T$; MC batch size $S$; Hutchinson probes $M$; Lanczos steps $L$; step $\eta$
**Output:** Variational parameters $\phi$
Initialize $\phi$
**repeat**
   Initialize $\widehat{\mathcal{L}} \leftarrow 0$
   **for** $s = 1$ **to** $S$ **do**
      Sample $\xi^{(s)} \sim \mathcal{N}(0, I_r)$
      $z^{(s)} \leftarrow s(f_\phi(\xi^{(s)}; \mathcal{C}))$
      $\Gamma^{(s)} \leftarrow \mathcal{S}(\exp(\text{mat}(Qz^{(s)})); \mu, \nu, T)$
      $E^{(s)} \leftarrow E(\Gamma^{(s)})$
      $\Delta_{ld}^{(s)} \leftarrow 0$
      **for** $m = 1$ **to** $M$ **do**
         Sample probe $\omega^{(m)} \in \{-1, 1\}^r$
         $v^{(0)} \leftarrow \omega^{(m)}/\|\omega^{(m)}\|$
         $T_L^{(s,m)} \leftarrow L$ step Lanczos on $G(z^{(s)})$ with $v^{(0)}$
         $\Delta_{ld}^{(s)} \leftarrow \Delta_{ld}^{(s)} + \|\omega^{(m)}\|^2 \, (e_1^\top \log(T_L^{(s,m)})e_1)$
      **end for**
      $\log \det \leftarrow \Delta_{ld}^{(s)}/M$
      $\ell^{(s)} \leftarrow -E^{(s)} + \log|\det J_{h_\phi}(\xi^{(s)})| + \frac{1}{2}\widehat{\log \det}$
      $\widehat{\mathcal{L}} \leftarrow \widehat{\mathcal{L}} + \ell^{(s)}$
   **end for**
   $\phi \leftarrow \phi + \eta \nabla_\phi(\widehat{\mathcal{L}}/S)$
**until** convergence

---

1989). We draw $M$ probe vectors $(\omega^{(1)}, \ldots, \omega^{(M)} \in \mathbb{R}^r)$ with $\mathbb{E}[\omega\omega^\top] = I$ (Rademacher ($\pm 1$) or standard Gaussian), and use the approximation

$$\text{Tr}(\log G(z)) \approx \frac{1}{M} \sum_{i=1}^{M} (\omega^{(i)})^\top \log(G(z)) \, \omega^{(i)}.$$

Finally, for each quadratic form, we use Stochastic Lanczos Quadrature (SLQ) (Ubaru et al., 2017; Chen et al., 2021). For each probe $\omega^{(i)}$, we run $L$ steps of Lanczos on $G(z)$ with starting vector

$$q_1^{(i)} = \frac{\omega^{(i)}}{\|\omega^{(i)}\|}.$$

This produces an $L \times L$ symmetric tridiagonal matrix $T_L^{(i)}(z)$. SLQ approximates

$$(\omega^{(i)})^\top \log(G(z)) \, \omega^{(i)} \approx \|\omega^{(i)}\|^2 \, e_1^\top \log(T_L^{(i)}(z)) \, e_1,$$

where $(e_1 = (1, 0, \ldots, 0)^\top \in \mathbb{R}^L)$. This requires only matrix-vector products with the Jacobian $J_{\bar{F}}$, which are efficiently computed via automatic differentiation. We prove in Appendix A that the coupling map and its associated Jacobian converge geometrically to their exact counterparts as $T \to \infty$. The time complexity scales with the size $n \times m$ as $\mathcal{O}(SnmTML)$.

# 5. Experiments

To ensure reproducibility, all architectural details, hyperparameters, and optimization settings are given in Appendix B.

## 5.1. Domain Adaptation on Toy Datasets

We evaluate SPVI on a cross-domain transport task between MNIST (source) and USPS (target), a benchmark setting in domain adaptation (DA) where OT is frequently employed to align distributions across varying acquisition domains (Courty et al., 2015; Redko et al., 2019).

### 5.1.1. ALEATORIC UNCERTAINTY MODELING

To study uncertainty stemming from intrinsic morphological variation, we analyze four semantically ambiguous pairs: $\{4 \to 9\}$, $\{3 \to 8\}$, $\{3 \to 7\}$, and $\{5 \to 9\}$. For any coupling $\Gamma \in \Pi(\mu, \nu)$, we define the morphed image via the barycentric projection $\mathcal{T}(\Gamma) \in \mathbb{R}^d$:

$$\mathcal{T}(\Gamma) := \text{diag}(\mu)^{-1}\Gamma \mathbf{Y}$$

where $\mathbf{Y} \in \mathbb{R}^{m \times d}$ is the matrix of target images, allowing us to visualize transport uncertainty directly for a source $i$, as shown in Figure 4. If $\Gamma$ maps source $i$ to target $j$, the resulting morph is the target image $y_j$. If $\Gamma$ splits the mass of $i$ across several targets, the morph is their weighted average.

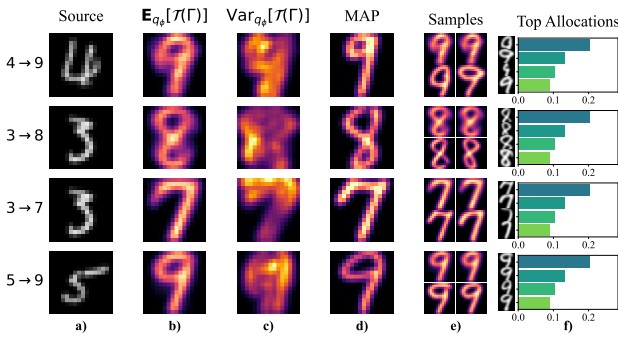

**Figure 4.** Each row shows transport from (a) a single MNIST source digit to USPS target digits. (b) Expected morph $\mathbb{E}_{q_\phi}[\mathcal{T}_i(\Gamma)]$, where $\mathcal{T}_i(\Gamma)$ is the $i$-th row of $\mathcal{T}(\Gamma)$; structural blur indicates regions of transport uncertainty. (c) Pixel-wise variance $\text{Var}_{q_\phi}[\mathcal{T}_i(\Gamma)]$, where brighter colors indicate higher transport ambiguity. (d) MAP on the mean cost (deterministic baseline). (e) Individual posterior samples. (f) Top four target indices by posterior mean mass.

### 5.1.2. MULTIMODALITY AND MIXED TARGETS

A key challenge in OT is when a single source mass can be plausibly mapped to distinct target clusters. We test our model's ability to resolve such ambiguities using three mixed target sets: $\{1\} \to \{3, 6\}$, $\{0\} \to \{2, 5, 9\}$, and $\{3\} \to \{4, 8\}$. As shown in Figure 5, standard MAP on the

mean cost selects only the lowest-cost mode. In contrast, our VI approach identifies distinct posterior modes and the posterior samples reflect the true diversity of potential mappings. These results demonstrate that SPVI identifies not

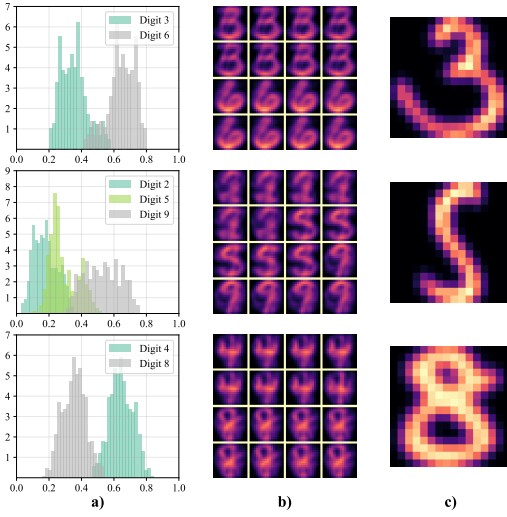

**Figure 5.** (a) Transport mass distribution across target digit classes shows multimodal posterior. (b) Posterior samples exhibit a diverse set of morphings favoring different digits. (c) Barycentric projection of the MAP on the mean cost (deterministic baseline).

just the most likely transport plan, but also the plausible alternatives. We now demonstrate the scalability of our framework in comparison to the sampling-based BayesOT.

### 5.2. High-Dimensional Scalability

We scale the problem from $m = n = 32$ to $n = 512$ using two Gaussian point clouds in $\mathbb{R}^2$ with squared Euclidean costs. We simulate uncertainty via $N = 20$ positive noisy realizations obtained from multiplicative log-normal perturbations of the base cost. We use a matrix-valued affine coupling flow, i.e. a RealNVP-style coupling architecture applied to the gauge-fixed logit matrix, using structured masks and convolutional conditioner networks (Dinh et al., 2015; Kingma & Dhariwal, 2018; Negri et al., 2023). As a gold-standard reference, we employ a HMC sampler (Mallasto et al., 2021) using the No-U-Turn Sampler (NUTS) (Hoffman & Gelman, 2014), and also add a comparison of MAP on the mean cost. Table 1 reports wall-clock runtime across scales. Posterior distance and cost-variance diagnostics are reported in Appendix B.2.

### 5.3. Joint Representation and Transport Learning

In many practical applications, the cost matrix is computed from learned features from raw data, and different feature extractors provide divergent views of the data. Consider a domain adaptation setting on the Office-31 dataset (Saenko et al., 2010) with a labeled source $\mathcal{D}_s = \{(x_i^s, y_i^s)\}_{i=1}^n$ and

**Table 1.** Runtime benchmark for Sinkhorn-VI against HMC across scales ($N = 20$ cost samples). MAP runtimes measure Sinkhorn on the mean cost with 100 iterations over 10 timing repeats. HMC runtimes for $n \leq 256$ use scale-dependent NUTS budgets (50 burn-in/100 samples at $n = 32$, 100/200 at $n = 64$, 200/500 at $n = 128$, and 500/1000 at $n = 256$) and are averaged over 10 runs; SPVI results are averaged over 5 runs. HMC chains are monitored via $\hat{R} < 1.1$ with a 10-hour wall clock and do not converge for $n > 256$. All experiments are run on CPU.

| Size ($n$) | Dim ($r$) | Method | Runtime (s) ↓ |
|---|---|---|---|
| $n = 32$ | $r = 961$ | MAP | $0.005 \pm 0.001$ |
| | | HMC | $61.6 \pm 23.0$ |
| | | **SPVI** | $59 \pm 8$ |
| $n = 64$ | $r = 3{,}969$ | MAP | $0.025 \pm 0.011$ |
| | | HMC | $329.0 \pm 34.2$ |
| | | **SPVI** | $157 \pm 32$ |
| $n = 128$ | $r = 16{,}129$ | MAP | $0.032 \pm 0.007$ |
| | | HMC | $1{,}217.1 \pm 238.9$ |
| | | **SPVI** | $580 \pm 116$ |
| $n = 256$ | $r = 65{,}025$ | MAP | $0.069 \pm 0.007$ |
| | | HMC | $21{,}533 \pm 3{,}200$ |
| | | **SPVI** | $2{,}144 \pm 429$ |
| $n = 512$ | $r = 261{,}121$ | MAP | $0.161 \pm 0.019$ |
| | | HMC | — |
| | | **SPVI** | $7{,}933 \pm 1{,}587$ |

an unlabeled target $\mathcal{D}_t = \{x_j^t\}_{j=1}^m$, where $y_i^s$ represents the class label among $K = 31$ possible classes. Define a classifier $g_\psi : \mathcal{X} \to \mathbb{R}^K$, and denote $g_{\psi,c}(x)$ as the logit for class $c$. We employ 4 frozen encoders: ResNet-50 (He et al., 2016), DeiT (Touvron et al., 2021), DINOv2 (Oquab et al., 2024), and OpenCLIP (Cherti et al., 2023) to construct cost matrices $\{C^{(k)}\}_{k=1}^4$ as

$$C_{ij}^{(k)}(\psi) = \alpha \tilde{D}_{ij}^{(k)} + \eta(t) L_{ij}(\psi)$$

with a tunable $\alpha$. Here $\tilde{D}_{ij}^{(k)}$ represents standardized $\ell_2$-normalized feature distances scaled by the matrix median to ensure that no view dominates purely by scale. The second term $L_{ij}(\psi)$ enforces label consistency using a one-vs-all squared hinge loss $\ell_{\mathrm{sh}}(y, s) = \max(0, 1 - ys)^2$:

$$L_{ij}(\psi) = \sum_{c=1}^K \ell_{\mathrm{sh}}\big(Y_{ic}^s, g_{\psi,c}(x_j^t)\big),$$

where $Y^s \in \{+1, -1\}^{n \times K}$ are the signed source labels. We optimize via block coordinate updates $(\phi^{(t)}, \psi^{(t)})$ over $T_{\mathrm{outer}}$ iterations.

(a) Update Transport ($\phi$): Given classifier $\psi^{(t)}$, update variational parameters $\phi$ to maximize $\mathcal{L}(\phi; \mathcal{C}(\psi^{(t)}))$.

(b) Update Classifier ($\psi$): Minimize the posterior-predictive risk. Let $\hat{P}(\Gamma) = \Gamma^\top P^s \in \mathbb{R}^{m \times K}$ be

the soft target labels induced by the coupling (where $P^s \in \{0,1\}^{n \times K}$ are one-hot source labels). We minimize:

$$\mathcal{J}(\psi) = \mathbb{E}_{\Gamma \sim q_\phi} \left[ \sum_{j=1}^{m} \sum_{c=1}^{K} \mathcal{L}_{\text{target}}(j, c, \psi, \Gamma) \right] + \lambda_s \mathcal{L}_{\text{source}}(\psi) + \lambda_\Omega \|\psi\|_2^2$$

where the target loss weighs positive and negative predictions by the transported mass:

$$\mathcal{L}_{\text{target}}(j, c, \psi, \Gamma) = \hat{P}_{jc}(\Gamma)\ell_{\text{sh}}(+1, g_{\psi,c}(x_j^t)) \quad (12)$$
$$+ (1 - \hat{P}_{jc}(\Gamma))\ell_{\text{sh}}(-1, g_{\psi,c}(x_j^t))$$

and the source loss

$$\mathcal{L}_{\text{source}}(\psi) = \sum_{i=1}^{n} \sum_{c=1}^{K} \ell_{\text{sh}}\left(Y_{ic}^s, g_{\psi,c}(x_i^s)\right). \quad (13)$$

To avoid early collapse, $\eta(t)$ linearly ramps from $0$ to $1$ over the outer iterations, gradually introducing the label constraint as the classifier improves. We compare our method against SurK (Zhang et al., 2013), a surrogate-kernel approach; OT-MM, which applies optimal transport with class-wise regularization optimized via a majorization-minimization algorithm (Chapel et al., 2021); and JDOT, which performs joint distribution optimal transport for domain adaptation (Courty et al., 2017). We also report a baseline (Base) obtained by training on the source domain and evaluating directly on the target domain. Since SPVI provides a distribution, we compute the MAP transport plan to compare the accuracy in Table 2. This demonstrates that,

**Table 2.** Classification accuracy (%) on Office-31 tasks.

| Domains | Base | SurK | OT-MM | JDOT | SPVI |
|---|---|---|---|---|---|
| amazon → webcam | 79.85 | 81.56 | **85.96** | 84.05 | 84.02 |
| amazon → dslr | 86.21 | **87.67** | 79.21 | 87.48 | 86.81 |
| webcam → amazon | 79.79 | 78.48 | **90.57** | 90.21 | 90.01 |
| webcam → dslr | 96.56 | 95.32 | **99.99** | 98.55 | 97.43 |
| dslr → amazon | 83.91 | 82.27 | **90.59** | 88.59 | 89.47 |
| dslr → webcam | 96.74 | 92.31 | 96.56 | 96.17 | **96.88** |
| Mean | 87.18 | 86.27 | 90.48 | **90.84** | 90.77 |

despite being designed to model a distribution over transport plans, SPVI achieves performance comparable to standard methods that estimate a single deterministic transport plan.

### 5.4. Single-Cell Fate Dynamics

Optimal transport has received considerable attention in recent years in single-cell biology, where it is used to map cell populations during differentiation measured by destructive experiments that prohibit one-to-one matching of cells (Schiebinger et al., 2019; Bunne et al., 2023b; 2024). We study fate uncertainty during neuronal differentiation in the Dentate Gyrus dataset for mouse hippocampus neurogenesis (Hochgerner et al., 2018). We treat the differentiation process as a cross-population transport problem between an early (source) and a late (target) cell population. Let $X \in \mathbb{R}^{n \times 30}$ and $Y \in \mathbb{R}^{m \times 30}$ denote the PCA embeddings of source and target cells, with rows $x_i^\top$ and $y_j^\top$ and uniform marginals, where $n = m = 300$. Interpreting source and target populations as snapshots of a biological process at two latent times, the coupling $\Gamma$ induces a probabilistic transition rule. We interpret the conditional distribution $\pi(j \mid i) := \Gamma_{ij}/\mu_i$ as a discrete approximation of the transition density $p(y \mid x_i)$ of a latent stochastic evolution operator (Zhang et al., 2025; Gu et al., 2025; Lavenant et al., 2024). We propagate uncertainty from the inferred developmental dynamics by constructing an ensemble of plausible transition cost matrices $\mathcal{C}$ by repeatedly reconstructing the cell-state geometry and the forward developmental transition structure under bootstrap perturbations of the single-cell data. Each cost matrix represents one plausible realization of the latent developmental dynamics between the source and target populations. See Appendix B.4 for details.

Our variational approximation $q_\phi(\Gamma \mid \mathcal{C})$ represents the uncertainty over this transition law. We capture this uncertainty through the posterior predictive variance, or *plasticity*, defined as

$$V_i := \sum_{\ell=1}^{30} \text{Var}_{q_\phi}([\mathcal{T}_i(\Gamma)]_\ell)$$

which serves as a proxy for branching uncertainty. As shown in Figure 6, multipotent progenitor cells exhibit higher Shannon entropy in their posterior mean fate distributions compared to committed lineages. Furthermore, plotting $V_i$ against developmental maturity reveals a funneling effect: early cells have less stable predicted destinations across plausible transition dynamics, whereas mature cells are transported consistently to a narrower region of the target population.

## 6. Conclusion and Future work

We have introduced a theoretically grounded framework for variational inference over the transport polytope. This allows us to learn a posterior distribution over possible transport plans and to assess uncertainty in matching on new samples. SPVI shows promising results on a large variety of tasks, including uncertainty-aware inference of single-cell fate dynamics, domain adaptation, and joint representation and transport learning.

SPVI is limited to balanced entropic optimal transport; extensions to unbalanced optimal transport would be highly valuable. Obtaining theoretical results on the optimization process under simple parametrizations of the normalizing flows would also be interesting. Lastly, we believe that

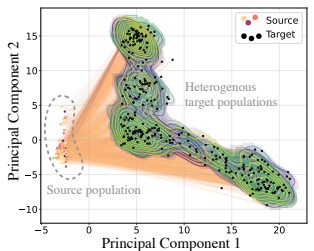 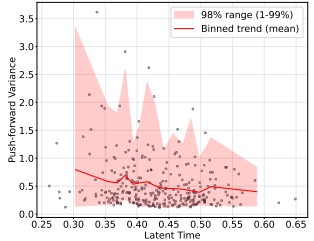

**Figure 6.** Left: Cells are mapped in PCA space. Source cells are colored by the Shannon entropy of the conditional fate distribution $\pi(j \mid i)$ derived from the posterior mean coupling. Transport lines are colored proportionally to their probability, connecting source cells to likely destinations. For the 5 most uncertain cells, density contours represent the variational posterior of barycentric fate mappings. Right: Relationship between $V_i$ and developmental maturity (latent time). Variance is computed across 200 variational posterior samples. The red trend line indicates mean variance across 15 quantile bins, with shaded bands showing the 1st to 99th percentile range.

further developments of generative methods for optimal transport plans that can reflect fundamental randomness in matching mechanisms are of high interest.

## Impact Statement

This paper introduces a scalable variational inference framework for Bayesian optimal transport that places expressive, reparameterized distributions directly on the interior of the transport polytope. By combining a gauge-fixed Sinkhorn parameterization with normalizing flows and a principled manifold density correction, our method replaces costly constrained MCMC with fast, differentiable optimization while retaining the ability to represent multimodal posteriors induced by uncertain or heterogeneous cost matrices. The approach enables uncertainty-aware transport plans at problem sizes where sampling-based BayesOT becomes impractical, and provides a general template for probabilistic inference over constrained couplings that can benefit downstream tasks such as robust matching, domain adaptation, and uncertainty-calibrated decision making.

## Acknowledgements

Funding from the Deutsche Forschungsgemeinschaft (DFG, German Research Foundation) within project 450330162 is gratefully acknowledged. LB's work at EPFL is supported by an EPFL AI Center Postdoctoral Fellowship.

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

## A. Truncated Sinkhorn Scaling

This appendix formalizes the use of a finite Sinkhorn iteration budget $T$ in our variational family and establishes uniform geometric convergence of the truncated coupling map.

**Setup.** Fix the compact latent set $Z := [-c, c]^r$ and define the gauge-fixed logit map $B(z) := \text{mat}(Qz) \in \mathcal{G}^\perp$, where $Q \in \mathbb{R}^{nm \times r}$ has orthonormal columns spanning $\text{vec}(\mathcal{G}^\perp)$. Let

$$K(z) := \exp(B(z)) \in \mathbb{R}^{n \times m}_{++}, \qquad \Psi(z) := \mathcal{S}(K(z); \mu, \nu) \in \mathcal{M},$$

denote the exact Sinkhorn coupling. Starting from $u^0 = \mathbf{1}_n$, define the truncated iterates

$$v^t(z) := \nu \oslash (K(z)^\top u^{t-1}(z)), \quad u^t(z) := \mu \oslash (K(z)v^t(z)), \quad \Psi_T(z) := \text{diag}(u^T(z))\, K(z)\, \text{diag}(v^T(z)). \quad (14)$$

We measure marginal feasibility via the residual vector

$$\text{res}_T(z) := \begin{bmatrix} \Psi_T(z)\mathbf{1}_m - \mu \\ \Psi_T(z)^\top \mathbf{1}_n - \nu \end{bmatrix} \in \mathbb{R}^{n+m}. \quad (15)$$

We first establish the following structural assumptions:

**A1.** The latent set $Z$ is compact and $Q$ is a fixed linear operator, ensuring a finite bound $b := \sup_{z \in Z} \|B(z)\|_\infty$.

**A2.** The energy $E$ extends to a Lipschitz function on a compact set $\mathcal{U} \subset \mathbb{R}^{n \times m}_{++}$ that contains $\{\Psi(z) : z \in Z\}$ and $\{\Psi_T(z) : z \in Z\}$ (for the fixed truncation $T$ used in the objective).

Under **A1**, $K(z) = \exp(B(z))$ is $C^2$ in $z$, with uniformly bounded first derivatives on $Z$, and all kernels satisfy $e^{-b} \leq K_{ij}(z) \leq e^b$ for all $(i, j)$ and $z \in Z$. We define the Hilbert projective metric for $x, y \in \mathbb{R}^d_{++}$ as $d_H(x, y) := \log(\max_i x_i/y_i) - \log(\min_i x_i/y_i)$. For $K \in \mathbb{R}^{n \times m}_{++}$ define the projective diameter $\Delta(K)$ and contraction coefficient $\kappa(K)$ as

$$\Delta(K) := \log \sup_{i,i',j,j'} \frac{K_{ij} K_{i'j'}}{K_{ij'} K_{i'j}}, \qquad \kappa(K) := \tanh(\Delta(K)/4).$$

Hence, **A1** implies the projective diameter satisfies $\Delta(K(z)) \leq 4b$ because:

$$\frac{K_{ij} K_{i'j'}}{K_{ij'} K_{i'j}} \leq \frac{e^b e^b}{e^{-b} e^{-b}} = e^{4b}.$$

**Theorem A.1.** *Assume A1. Then there exists a rate*

$$\rho := \sup_{z \in Z} \kappa(K(z))^2 \leq \tanh(b)^2 < 1$$

*and a constant $C_\Gamma < \infty$ (depending only on $b, \mu, \nu$) such that for all $z \in Z$ and $T \geq 1$,*

$$\|\Psi_T(z) - \Psi(z)\|_1 \leq C_\Gamma \rho^{T-1}, \qquad \|\text{res}_T(z)\|_1 \leq C_\Gamma \rho^{T-1}. \quad (16)$$

*Proof.* The Birkhoff–Hopf theorem implies that the projective iterates contract: the full Sinkhorn step (one $v$ update followed by one $u$ update) is a contraction in $d_H$ with coefficient at most $\kappa(K(z))^2$ (Franklin & Lorenz, 1989; Idel, 2016). Under **A1**, this yields a uniform rate $\rho < 1$ on $Z$. To translate contraction in $d_H$ into an $\ell^1$ bound for $\Psi_T$, we use that the iterates $(u^t, v^t)$ remain in a forward-invariant, projectively bounded set determined by $(b, \mu, \nu)$ (e.g., $d_H(u^t, \mathbf{1}_n)$ is uniformly bounded). On such a compact log-slice, $d_H$ controls $\|\log u - \log(\alpha u')\|_\infty$ for an appropriate scalar $\alpha$. Since $(u, v) \mapsto \text{diag}(u)K\text{diag}(v)$ is smooth and hence Lipschitz on this compact set, the contraction transfers to $\|\Psi_T - \Psi\|_1$ with a uniform constant $C_\Gamma$. Note that by construction $\Psi_T(z)\mathbf{1}_m = \mu$, hence the residual reduces to the column error $\|\Psi_T(z)^\top \mathbf{1}_n - \nu\|_1$, which decays geometrically under the same projective contraction. $\square$

Now according to **A2**, let $E : \mathcal{M} \to \mathbb{R}$ be Lipschitz on the compact subset $\{\Psi(z) : z \in Z\} \subset \mathcal{M}$. Then Theorem A.1 implies

$$\sup_{z \in Z} |E(\Psi_T(z)) - E(\Psi(z))| \leq \text{Lip}(E) \sup_{z \in Z} \|\Psi_T(z) - \Psi(z)\|_1 = \mathcal{O}(\rho^T).$$

**Lemma A.2.** *Suppose* $\Phi : Z \times \mathcal{X} \to \mathcal{X}$ *is* $C^1$ *and for all* $z \in Z$, $x \in \mathcal{X}$, $\|D_x\Phi_z(x)\| \leq \eta < 1$ *and* $\sup_{z,x} \|D_z\Phi_z(x)\| < \infty$. *Let* $x^t(z)$ *be defined by* $x^t(z) = \Phi_z(x^{t-1}(z))$ *with* $x^0$ *independent of* $z$. *Then the fixed point* $x^*(z)$ *is* $C^1$, *and the derivative iterates* $Dx^t(z)$ *converge to* $Dx^*(z)$ *geometrically and uniformly:*

$$\sup_{z \in Z} \|Dx^t(z) - Dx^*(z)\| \;\leq\; C\,\eta^t,$$

*where* $C$ *depends on* $\sup \|D_z\Phi_z\|$ *and* $(1 - \eta)^{-1}$.

We now address convergence of the Jacobian

$$J_T(z) := \nabla_z \operatorname{vec}(\Psi_T(z)) \in \mathbb{R}^{nm \times r}, \qquad J_\infty(z) := \nabla_z \operatorname{vec}(\Psi(z)).$$

Uniform geometric convergence of $J_T$ is not implied by Theorem A.1 alone; it additionally requires stability of the *linearization* of Sinkhorn under truncation. We state a sufficient condition that can be verified under bounded logits and strictly positive marginals.

Let $\alpha^t(z) := \log u^t(z) \in \mathbb{R}^n$ and $\beta^t(z) := \log v^t(z) \in \mathbb{R}^m$. Fix the 1-dimensional scaling ambiguity by imposing a gauge, e.g. $\langle \mu, \alpha \rangle = 0$ at every step. Let $x^t(z)$ denote the resulting gauge-fixed log-scaling state (e.g. $x^t = \alpha^t$). Write the full Sinkhorn step as a map $x^t = \Phi_z(x^{t-1})$.

**A3.** There exists a norm $\|\cdot\|$ on the gauge-fixed state space, a compact forward-invariant set $\mathcal{X}$, and constants $\eta \in (0,1)$ and $L_z < \infty$ such that for all $z \in Z$ and all $x \in \mathcal{X}$:

$$\|D_x\Phi_z(x)\| \leq \eta, \qquad \|D_z\Phi_z(x)\| \leq L_z.$$

Moreover, $\Phi_z$ is $C^1$ jointly in $(x, z)$ on $\mathcal{X} \times Z$.

**Theorem A.3.** *Assume A1 and A3. Then: (i) the exact map* $\Psi : Z \to \mathcal{M}$ *is* $C^1$, *(ii) the truncated Jacobians converge uniformly at a geometric rate: there exist constants* $C_J < \infty$ *and* $\tilde{\rho} \in (0,1)$ *such that for all* $T \geq 1$,

$$\sup_{z \in Z} \|J_T(z) - J_\infty(z)\| \;\leq\; C_J\,\tilde{\rho}^{T-1}, \tag{17}$$

*where* $\tilde{\rho}$ *can be taken as* $\eta$ *(up to norm equivalence constants induced by the gauge).*

*Proof.* Under **A3**, the fixed point $x^*(z)$ of $x = \Phi_z(x)$ exists uniquely on $\mathcal{X}$ and depends smoothly on $z$ by the implicit function theorem / contraction mapping theorem for $C^1$ contractions. Differentiating the recursion yields

$$D_z x^t(z) = D_x\Phi_z(x^{t-1}(z))\, D_z x^{t-1}(z) + D_z\Phi_z(x^{t-1}(z)).$$

Subtracting the fixed-point derivative equation and using $\|D_x\Phi_z\| \leq \eta < 1$ gives a linear contraction in the derivative error, hence $\sup_z \|D_z x^t(z) - D_z x^*(z)\| = \mathcal{O}(\eta^t)$ uniformly. Finally, $\Psi_T(z)$ is a smooth composition of $(u^T(z), v^T(z), K(z))$ with $(u, v, K) \mapsto \operatorname{diag}(u) K \operatorname{diag}(v)$, so convergence of derivatives of the scaling state implies (17). $\square$

# B. Implementation details

The following implementation details are the same across all experiments unless stated otherwise.

**Construction of $Q$.** We do not form $Q \in \mathbb{R}^{nm \times r}$ but implement $B = \operatorname{mat}(Qz) \in \mathcal{G}^\perp$ via an explicit factorization. We precompute the QR bases

$$A_n = [e_1 - e_n, \ldots, e_{n-1} - e_n] = U R_n,$$
$$A_m = [e_1 - e_m, \ldots, e_{m-1} - e_m] = V R_m,$$

so $U \in \mathbb{R}^{n \times (n-1)}$, $V \in \mathbb{R}^{m \times (m-1)}$ have orthonormal columns and satisfy $U^\top \mathbf{1}_n = 0$, $V^\top \mathbf{1}_m = 0$. We can now choose *any* $z \in \mathbb{R}^r$, reshaped to form $S = \operatorname{reshape}(z, (n-1, m-1))$ and set $B := U S V^\top$, which enforces $B\mathbf{1}_m = 0$ and $B^\top \mathbf{1}_n = 0$ by construction. This is equivalent to using $Q = V \otimes U$ but avoids storing $Q$.

**Hyperparameters.** We fix the number of Sinkhorn iterations $T$ such that the column marginal residual is below $10^{-10}$ for a reasonable $n$. Typically, this happens within $T \leq 20$ steps for $n \leq 300$. We use $S = 4$ Monte Carlo samples for the expectation over $\xi$. For the trace estimator of the Gram logarithm, we use $M = 8$ Rademacher probe vectors and $L = 10$ Lanczos steps. To choose $c$, we use the robust lower-bound heuristic $c \geq \frac{\max(C) - \min(C)}{2\varepsilon}$ and set $c = 1.2 \times \frac{\max(C) - \min(C)}{2\varepsilon}$. To ensure numerical stability at larger scales, all Sinkhorn operations and variational density evaluations were performed in the log-domain using the log-sum-exp trick.

**Optimization details.** We use Adam with learning rate $\eta = 10^{-3}$, $(\beta_1, \beta_2) = (0.9, 0.999)$, and $\epsilon_{\text{ADAM}} = 10^{-8}$. We apply gradient clipping with maximum global norm $1.0$ to stabilize training. We use a linear warmup for the first $T_{\text{warm}}$ iterations up to the base learning rate, followed by cosine decay down to a minimum learning rate of $10^{-5}$:

$$
\eta_t = \begin{cases}
\eta \cdot \dfrac{t}{T_{\text{warm}}}, & t \leq T_{\text{warm}}, \\[2mm]
\eta_{\min} + \dfrac{1}{2}(\eta - \eta_{\min})\left(1 + \cos\left(\pi \dfrac{t - T_{\text{warm}}}{T_{\max} - T_{\text{warm}}}\right)\right), & T_{\text{warm}} < t \leq T_{\max},
\end{cases}
$$

with $\eta_{\min} = 10^{-5}$, $T_{\max} = 200$ the total number of training iterations, and $T_{\text{warm}} = 20$. We use early stopping and terminate training if the ELBO does not improve for 50 consecutive steps.

### B.1. Domain Adaptation on Toy Datasets

We utilize the MNIST and USPS handwritten digit datasets. All images are resized to $16 \times 16$ pixels and pixel values are normalized to $[0, 1]$. For both experiments, we randomly sample $n = 25$ source images and $m = 25$ target images. The variational distribution is parameterized by a RealNVP normalizing flow consisting of 2 coupling layers. Within each coupling layer, the translation and scale networks are implemented as 2-layer MLPs with 32 hidden units and ReLU activations.

### B.2. High-Dimensional Scalability

We generate synthetic 2D point clouds drawn from Gaussian mixtures. The base cost matrix $C$ is the squared Euclidean distance. We simulate cost uncertainty by generating $N = 20$ positive noisy realizations

$$
C_{ij}^{(k)} = C_{ij} \exp\left(\sigma Z_{ij}^{(k)} - \frac{1}{2}\sigma^2\right), \qquad Z_{ij}^{(k)} \sim \mathcal{N}(0, 1),
$$

with noise level $\sigma = 0.1$. The centering term keeps $\mathbb{E}[C_{ij}^{(k)} \mid C_{ij}] = C_{ij}$ while preserving nonnegativity. We compare SPVI against the BayesOT implementation (Mallasto et al., 2021) using the No-U-Turn Sampler (NUTS) (Hoffman & Gelman, 2014). Chains are warm-started using the deterministic Sinkhorn solution on the mean cost matrix. For the HMC runtime benchmark ($n \leq 256$), we use scale-dependent NUTS budgets: 50 burn-in/100 samples at $n = 32$, 100/200 at $n = 64$, 200/500 at $n = 128$, and 500/1000 at $n = 256$. We average over 10 runs with a target acceptance rate of $0.8$. Chain convergence was assessed using the $\hat{R}$ diagnostic, with a threshold of $\hat{R} < 1.1$ for all parameters and a 10-hour wall-clock limit. As the dimensionality of the transport polytope grows quadratically with the support size, we scale the capacity of the architecture and the HMC sampling budget accordingly, as shown in Table 4. The conditioner for SPVI uses 2 residual conv blocks and $3 \times 3$ kernels for all cases.

**Posterior sample distance.** Let $\{\Gamma_i^{\text{VI}}\}_{i=1}^{S}$ and $\{\Gamma_j^{\text{HMC}}\}_{j=1}^{S}$ denote equal-size posterior sample sets, and let $g_i^{\text{VI}} = \text{vec}(\Gamma_i^{\text{VI}})$ and $g_j^{\text{HMC}} = \text{vec}(\Gamma_j^{\text{HMC}})$. For $n \leq 128$, we report the empirical 2-Wasserstein distance

$$
\widehat{\mathcal{W}}_2 = \left[\min_{\pi \in \Pi_S} \frac{1}{S} \sum_{i=1}^{S} \left\| g_i^{\text{VI}} - g_{\pi(i)}^{\text{HMC}} \right\|_2^2\right]^{1/2},
$$

where $\Pi_S$ is the set of permutations of $\{1, \ldots, S\}$ with $S = 200$. For $n > 128$, exact empirical matching is expensive because $g_i \in \mathbb{R}^{n^2}$, so we report the sliced 2-Wasserstein distance

$$
\widehat{\mathcal{SW}}_2 = \left[\frac{1}{K} \sum_{k=1}^{K} \mathcal{W}_2^2 \left(\frac{1}{S} \sum_{i=1}^{S} \delta_{\theta_k^\top g_i^{\text{VI}}}, \frac{1}{S} \sum_{j=1}^{S} \delta_{\theta_k^\top g_j^{\text{HMC}}}\right)\right]^{1/2},
$$

with random directions $\theta_k \sim \text{Unif}(\mathbb{S}^{n^2-1})$. In one dimension, each inner $\mathcal{W}_2$ is computed by sorting the projected samples.

**Table 3.** Posterior quality diagnostics for SPVI in the high-dimensional scalability experiment. Metrics are averaged over 5 runs. Variance is computed using 200 samples.

| **Size** $(n)$ | $r = (n-1)^2$ | $\mathcal{W}_2/\mathcal{SW}_2 \downarrow$ | $\text{Var}[\langle C, \Gamma \rangle]$ |
|---|---|---|---|
| $n = 32$ | 961 | $0.00248 \pm 0.00060$ | $0.00169 \pm 0.00009$ |
| $n = 64$ | 3,969 | $0.00261 \pm 0.00055$ | $0.00856 \pm 0.00033$ |
| $n = 128$ | 16,129 | $0.00169 \pm 0.00041$ | $0.00930 \pm 0.00042$ |
| $n = 256$ | 65,025 | $0.00218 \pm 0.00044$ | $0.00820 \pm 0.00061$ |
| $n = 512$ | 261,121 | — | $0.00861 \pm 0.00015$ |

**Table 4.** SPVI architecture and HMC hyperparameters across scales.

| $n$ | $r$ | Layers | Hidden channels | Burn-in | NUTS samples |
|---|---|---|---|---|---|
| 32 | 961 | 2 | 16 | 50 | 100 |
| 64 | 3,969 | 4 | 32 | 100 | 200 |
| 128 | 16,129 | 4 | 32 | 200 | 500 |
| 256 | 65,025 | 8 | 32 | 500 | 1,000 |
| 512 | 261,121 | 8 | 32 | 1,000 | 2,000 |

## B.3. Joint Representation and Transport Learning

We utilize an ensemble of four frozen pre-trained encoders to generate diverse cost views. For each encoder, the input images are resized to $224 \times 224$ and normalized using ImageNet statistics. For each view $k$, we compute pairwise distances between source and target embeddings:

$$D_{ij}^{(k)} = d\big(r^{(k)}(x_i^s),\ r^{(k)}(x_j^t)\big).$$

where

$$d(u, v) = \|u - v\|^2.$$

Distances from different encoders have very different scales, so we normalize each view:

$$\tilde{D}_{ij}^{(k)} = \frac{D_{ij}^{(k)}}{\text{median}(D^{(k)}) + \epsilon}.$$

The resulting $\tilde{D}^{(k)}$ are comparable across $k$. We solve the block coordinate descent problem using $T_{\text{outer}} = 30$ iterations and $T = 40$ for Sinkhorn. We use the Adam optimizer for both $\phi$ and $\psi$ with learning rates $\eta_\phi = 10^{-3}$ and $\eta_\psi = 10^{-3}$, respectively. We search for the best $\alpha \in \{10^{-5}, 10^{-4}, 10^{-3} \ldots, 1\}$, $\lambda_s \in \{10^{-3}, \ldots, 1, 1.5\}$, and weight decay $\lambda_\Omega \in \{10^{-4}, \ldots, 1\}$. The label consistency weight $\eta(t)$ ramps linearly from 0 to 1 over $T_{\text{ramp}} = 0.5 \times T_{\text{outer}}$. We pick a single classifier feature map $\varphi(x)$, then

$$g_\psi(x) = W\varphi(x) + b, \qquad \psi = (W, b), \quad W \in \mathbb{R}^{K \times d},\ b \in \mathbb{R}^K,$$

and $g_{\psi,c}(x) = w_c^\top \varphi(x) + b_c$. The squared hinge then uses these margins directly:

$$\ell_{\text{sh}}(Y_{ic}^s, g_{\psi,c}(x)) = \max(0, 1 - Y_{ic}^s, g_{\psi,c}(x))^2.$$

Because the label term $L_{ij}(\psi)$ is **shared across views** $k$, $g_\psi$ should *not* depend on $k$. To define one shared $\varphi(x)$ from the same frozen encoders, we concatenate their normalized embeddings:

Let $r^{(k)}(x)$ be the frozen embedding from encoder $k$, and $\bar{r}^{(k)}(x) = r^{(k)}(x)/|r^{(k)}(x)|_2$. Define

$$\varphi(x) = \big[\bar{r}^{(1)}(x) \mid \bar{r}^{(2)}(x) \mid \bar{r}^{(3)}(x) \mid \bar{r}^{(4)}(x)\big] \in \mathbb{R}^{d_1+d_2+d_3+d_4}.$$

We warm start the classifier for 5 epochs. For SurK, we search $d \in \{1, 4, 7, \ldots, 31\}$. For OT-MM, we search the regularization parameter $\eta \in \{1, \ldots, 10^5\}$. We use a linear-kernel SVM for classification for each method.

## B.4. Single-Cell Fate Dynamics

We use the Dentate Gyrus scRNA-seq dataset from Hochgerner et al. (2018). Cells are represented by the top 30 principal components ($d = 30$) of the top 2,000 highly variable genes. We sample $n = 300$ cells from the earliest latent-time quantile ($\leq 15\%$ of the latent time) as source and $m = 300$ cells from the latest latent-time quantile ($\geq 85\%$ of the latent time) as target. To construct the cost ensemble $\mathcal{C}$, we generate $N = 50$ subsampling realizations of the inferred developmental dynamics. For each realization, we subsample $80\%$ of the highly variable genes without replacement, recompute the 30-dimensional PCA embedding, and construct a 30-nearest-neighbor graph in the resulting cell-state space. We orient the local transition structure forward along latent time and compute source-to-target transition costs from the resulting forward developmental graph. We compute $L$-step forward diffusion probabilities $P_{ij}^{(k)}$ from source cells to target cells and define $C_{ij}^{(k)} = -\log(P_{ij}^{(k)} + \delta)$, with $\delta = 10^{-8}$. Edges are reweighted by a forward-time kernel and row-normalized to obtain a Markov transition matrix. This gives an ensemble of plausible transition cost matrices $\mathcal{C} = \{C^{(1)}, \dots, C^{(50)}\}$, reflecting uncertainty in the inferred developmental dynamics rather than perturbations of the transport plans themselves.

The variational posterior $q_\phi(\Gamma|\mathcal{C})$ is parameterized by an 8-layer RealNVP flow. The plasticity score $V_i$ is computed using $S = 200$ posterior samples from the trained flow. For each posterior sample $\Gamma^{(s)}$, we compute the barycentric fate prediction

$$\mathcal{T}_i(\Gamma^{(s)}) = \sum_{j=1}^{m} \frac{\Gamma_{ij}^{(s)}}{\mu_i} y_j \in \mathbb{R}^{30}.$$

The plasticity score is then estimated as

$$V_i = \frac{1}{S-1} \sum_{s=1}^{S} \sum_{\ell=1}^{30} \left( [\mathcal{T}_i(\Gamma^{(s)})]_\ell - \overline{[\mathcal{T}_i(\Gamma)]_\ell} \right)^2,$$

where

$$\overline{[\mathcal{T}_i(\Gamma)]_\ell} = \frac{1}{S} \sum_{s=1}^{S} [\mathcal{T}_i(\Gamma^{(s)})]_\ell.$$

## C. Impact of Sinkhorn Iterations

For most experiments, we find that $T \leq 20$ results in a marginal violation of $10^{-10}$. To isolate the contributions of our architectural choices, we conduct controlled experiments focusing on the Sinkhorn depth, approximation fidelity, and the expressivity of the variational family. Throughout these trials, we fix the entropic regularization $\epsilon = 0.05$, the problem size $n = 256$, and the noise level $\sigma = 0.1$ to ensure consistency across comparisons. We vary $T \in \{5, 10, 20, 50\}$ while maintaining a fixed budget of $S = 4$ Monte Carlo samples, $L = 10$ Lanczos steps, and $M = 8$ probes over 200 iterations. Results are averaged over 5 runs.

## D. Proof of Proposition 4.2

**Proposition D.1** (Global Diffeomorphism of the Gauge-Fixed Map). *Let $\mathcal{G} = \{a\mathbf{1}_m^\top + \mathbf{1}_n b^\top : a \in \mathbb{R}^n, b \in \mathbb{R}^m\}$ be the subspace of additive gauge freedoms, and let $\mathcal{G}^\perp$ be its orthogonal complement in $\mathbb{R}^{n \times m}$. Let $Q \in \mathbb{R}^{nm \times r}$ be a matrix with orthonormal columns spanning $\mathrm{vec}(\mathcal{G}^\perp)$. Define the map $\Psi : \mathbb{R}^r \to \mathcal{M}$ by*

$$\Psi(z) := \mathcal{S}\big(\exp(\mathrm{mat}(Qz)); \mu, \nu\big). \tag{18}$$

*Then $\Psi$ is a smooth global diffeomorphism from $\mathbb{R}^r$ onto $\mathcal{M}$. In particular, it is globally injective and its Jacobian $J_\Psi(z)$ has full column rank $r$ everywhere.*

*Proof.* Let $\mathcal{S}_{\log} : \mathbb{R}^{n \times m} \to \mathcal{M}$ denote the log-domain Sinkhorn map $\mathcal{S}_{\log}(B) = \mathcal{S}(\exp(B); \mu, \nu)$. By the uniqueness of Sinkhorn scaling factors (Peyré & Cuturi, 2019), for any $\Gamma \in \mathcal{M}$, the kernel $K = \exp(B)$ such that $\mathcal{S}(K) = \Gamma$ is unique up to left and right diagonal scaling. In the log-domain, this implies $\mathcal{S}_{\log}(B) = \mathcal{S}_{\log}(B')$ if and only if $B - B' \in \mathcal{G}$, where $\mathcal{G}$ is the subspace of gauge directions. Thus, the fibers are affine subspaces:

$$\mathcal{S}_{\log}^{-1}(\Gamma) = \log \Gamma + \mathcal{G}. \tag{19}$$

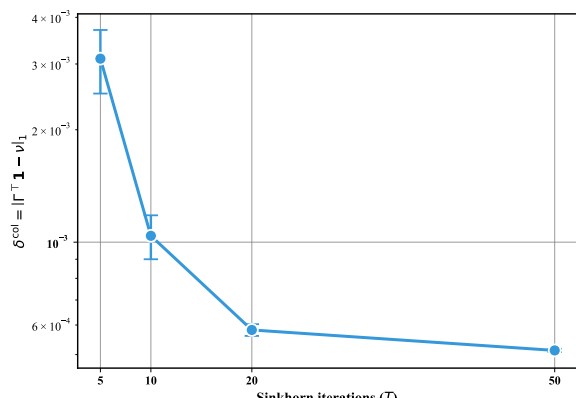 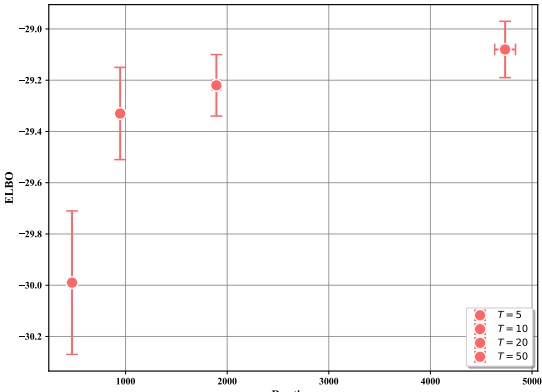

**Figure 7.** (a) Column error vs. Sinkhorn iterations $T$ shows approximately geometric decay until hitting a precision floor near $T = 50$. (b) ELBO vs. runtime for different Sinkhorn depths $T$. Higher $T$ improves performance at increased computational cost.

The domain of $\Psi$ is the subspace $\mathcal{G}^\perp = \text{im}(Q)$. Since $\mathbb{R}^{n \times m} = \mathcal{G} \oplus \mathcal{G}^\perp$ is an orthogonal direct sum, every affine translate of $\mathcal{G}$ intersects $\mathcal{G}^\perp$ at exactly one point. Specifically, for any $\Gamma \in \mathcal{M}$, the unique $z \in \mathbb{R}^r$ satisfying $\Psi(z) = \Gamma$ is given by the orthogonal projection of $\log \Gamma$ onto $\mathcal{G}^\perp$ expressed in the $Q$-basis:

$$\Psi^{-1}(\Gamma) = Q^\top \text{vec}\big(P_{\mathcal{G}^\perp}(\log \Gamma)\big). \tag{20}$$

The existence of this explicit inverse map $\Psi^{-1} : \mathcal{M} \to \mathbb{R}^r$ establishes global bijectivity. The map $\Psi$ is a composition of smooth maps: the linear embedding $Q$, the entrywise exponential, and the Sinkhorn operator. The Sinkhorn operator $\mathcal{S}(\cdot; \mu, \nu)$ is smooth on the positive orthant $\mathbb{R}_{++}^{n \times m}$ as a consequence of the Implicit Function Theorem applied to the marginal constraints (Knight, 2007). Similarly, $\Psi^{-1}$ is smooth on $\mathcal{M}$ as it is a composition of the smooth entrywise logarithm and linear projections. Since $\Psi$ and $\Psi^{-1}$ are both smooth, $\Psi$ is a global diffeomorphism. Consequently, the differential $d\Psi_z$ is an isomorphism between the tangent spaces $T_z \mathbb{R}^r$ and $T_{\Psi(z)} \mathcal{M}$, ensuring that $J_\Psi(z)$ has full column rank $r$ for all $z \in \mathbb{R}^r$. $\qquad \square$

