# OpenReview forum: "Variational Inference for Uncertain Optimal Transport via Sinkhorn Parametrization"
_ICML.cc/2026/Conference — ICML 2026 regular_

### Official Review · Reviewer_oDYV · 2026-03-09

**Soundness:** 3
**Presentation:** 2
**Significance:** 3
**Originality:** 3
**Overall Recommendation:** 5
**Confidence:** 4

**Summary:**

This paper proposes a method for "sampling" from collections of entropic couplings: Given multiple cost matrices, their goal is to sample from a distribution over couplings (for the purposes of uncertainty quantification, for example). To do so, the authors first appropriately define a Gibbs (posterior) measure over the space of couplings based on the log-sup-exp formula as well as a prior on the couplings. To ultimately sample from this posterior, they leverage (among a series of numerical manipulations) the differentiability of Sinkhorn iterates combined with existing generative models (e.g., normalizing flows) to optimize over a variational objective.

**Compliance With Llm Reviewing Policy:**

Affirmed.

**Final Justification:**

Addressed my concerns

**Key Questions For Authors:**

- What do the authors mean by the "truncated Sinkhorn operator"? (introduction)
- Can the authors please elaborate their experimental details? In Section 5.1.1, what is the considered dataset? Is it all sets of the digit $4$ and all sets of the digit $9$ (for example)? If so, what is the cost and what are the couplings?
- While the method is clearly better than e.g. NUTS, it still appears to be quite slow. Are there any "dream" or "ideal" aspects to the algorithm that would make this faster?

Some typos:
- C2 L58 "machine learning **and** has been embraced..." (or possibly restructure the sentence)
- C2 L419 "allows to **learn** a posterior... "
-

**Limitations:**

yes

**Strengths And Weaknesses:**

The method is sound and appears to be original, and certainly requires quite a bit of ingenuity to make use of all the relevant parts to provide a working algorithm. Indeed, one can feasibly construct a Gibbs posterior as prescribed and then optimize over a set, and the use of the differentiability of Sinkhorn solvers is one approach. The significance of the work is hard to determine, however, as the experiments are poorly explained---even in section 5.1.1, I do not understand what the experiments are trying to do---and the motivation is not entirely clear. For my personal tastes, there's also no result guaranteeing that this proposed methodology is guaranteed to sample from what they claim (unless this is perhaps "obvious").

---

> ### Author Rebuttal · Authors · 2026-03-29
>
> We thank the reviewer for the thoughtful reading and for identifying several places where the presentation was too compressed. The reviewer is right that our wording should more clearly distinguish between exact posterior sampling and sampling from a learned variational approximation.
>
> **On whether the method is guaranteed to sample from the claimed posterior.**
>
> Assuming we understood the concern correctly, the reviewer is right that the method does not produce exact posterior samples in the MCMC sense. SPVI learns a variational approximation $q_\phi(\Gamma)$ to the Gibbs posterior and then samples from $q_\phi$. So the correct claim is that the method performs **variational posterior approximation**, not exact posterior simulation. However flexible models like RealNVP are good approximators, as can be seen from the low $W_2$ values in Table 1. We will revise the wording accordingly in the introduction and experimental sections to emphasize this.
>
> **On the meaning of the truncated Sinkhorn operator.**
>
> By the “truncated Sinkhorn operator” we mean the map obtained after only $T$ alternating scaling steps, as opposed to the exact Sinkhorn fixed point. If $\Psi(z)=\mathcal S(\exp(B(z));\mu,\nu)$ denotes the exact coupling map, then $\Psi_T(z)$ is its finite-iteration approximation. This distinction matters because $\Psi_T$, and not $\Psi$, is the object actually used and differentiated through in the algorithm. In the appendix A, we basically show that this practical approximation remains well controlled.
>
> **On the toy digit experiments in Section 5.1.**
>
> Each toy experiment is a fully finite OT problem between two explicit image collections. We use the MNIST and USPS handwritten digit datasets, resize all images to $16\times16$, and normalize pixel values to $[0,1]$, so each support point lies in $\mathbb R^{256}$. In both the pairwise and mixed-target experiments, we randomly sample $n=25$ source images and $m=25$ target images, with uniform marginals $\mu_i=\nu_j=1/25$. Thus each coupling $\Gamma\in\Pi(\mu,\nu)$ is a $25\times25$ matrix, and the conditional distribution for source image (i) is $\pi(j\mid i)=\Gamma_{ij}/\mu_i=25\Gamma_{ij}$. The base cost matrix is computed from the flattened normalized images using squared Euclidean distance. The barycentric projection therefore maps each source image to a weighted average of the $25$ target images in $\mathbb R^{256}$. In experiments such as $4\to 9$, the source support consists of $25$ MNIST images of the digit $4$ and the target support consists of $25$ USPS images of the digit $9$; in mixed-target experiments such as $1\to{3,6}$, the target set is drawn from the union of the indicated target classes in order to induce genuine multimodal alignment structure. We will move these details from the appendix into the main text to make the experimental setup more transparent.
>
> **On possible “dream” directions for making the method faster.**
>
> This is a very interesting question, and one we have also been thinking about quite a lot. We agree with the reviewer that, although SPVI is substantially faster than NUTS/HMC, it is still not “cheap” in an absolute sense. In our view, this is partly the cost of insisting on a geometrically faithful variational distribution on the interior of the transport polytope, rather than settling for a faster but less structured approximation. The expensive part is not sampling from the flow itself, but accounting for the local geometry of the Sinkhorn chart through the manifold log-volume term $\frac12 \log\det(J^\top J)$.
>
> The direction we find most promising as a next version of this method is to preserve the same geometric picture while replacing backpropagation through many Sinkhorn iterations by implicit differentiation through the Sinkhorn fixed point [1]. This is attractive because the dominant cost comes from repeated Jacobian-vector products inside the stochastic log-volume estimator. At the same time, this is not simply an implementation detail: it would require a separate treatment of the gauge-fixed fixed-point equations and a different derivative analysis. The difficulty is that we do not just need to differentiate the fixed point once; we would need an efficient and numerically stable implicit-differentiation procedure for many repeated Jacobian actions inside the SLQ estimator. For this paper, we chose the more explicit route where the forward map, the backward pass, and the approximation analysis all refer to the same truncated Sinkhorn object. We therefore see the current method as a first principled construction in this direction, rather than the final word on computational efficiency.
>
> **On typos and wording.**
> Thank you; we will correct the noted typos.
>
> References:
>
> [1] Scieur et al., “The Curse of Unrolling”, NeurIPS 2023

---

> > ### Author Rebuttal · Reviewer_oDYV · 2026-04-03
> >
> > Thank you for the response, I will update my score.

---

### Official Review · Reviewer_CeoH · 2026-03-12

**Soundness:** 3
**Presentation:** 3
**Significance:** 3
**Originality:** 3
**Overall Recommendation:** 5
**Confidence:** 2

**Summary:**

The paper addresses the problem of performing posterior inference over optimal transport plans when the ground cost is uncertain or stochastic. It introduces a novel scalable variational framework, Sinkhorn-parameterized Variational Inference (SPVI). The framework includes three main components: i) a gauge-fixed Sinkhorn reparameterization that maps unconstrained latent variables onto the transport polytope while respecting marginal constraints, ii) a normalizing flow to model flexible distributions over transport plans, and iii) a tractable ELBO objective computed via Hutchinson trace estimation and Stochastic Lanczos Quadrature. Experiments show that SPVI demonstrates strong empirical performance across three distinct evaluation settings including scalability, domain adaptation and single-cell fate dynamics .

**Compliance With Llm Reviewing Policy:**

Affirmed.

**Key Questions For Authors:**

$\bullet$ The paper uses W2 distance as a primary evaluation metric in Table 1 to benchmark SPVI against HMC and MAP. how is this metric computed?

**Limitations:**

Although SPVI achieves a substantial speedup over HMC, its computational cost remains high in absolute terms. As reported in Table 1, SPVI requires approximately 25,083 seconds (roughly 7 hours) at n=2048, which, while superior to HMC, is still prohibitive for real-world deployment scenarios where low latency or repeated inference is required.

**Strengths And Weaknesses:**

## Strengths

$\bullet$ The paper presents a novel differentiable reparameterization technique for the transport polytope, enabling normalizing flows to model transport plans while satisfying marginal constraints, thereby making variational inference (VI) for optimal transport (OT) feasible and scalable.

$\bullet$ The paper provides rigorous theoretical justification for the use of finite-step Sinkhorn iterations in the variational objective.

$\bullet$ While maintaining comparable posterior approximation quality, SPVI achieves roughly a 4-5× speedup over HMC across all tested scales.

## Weaknesses

$\bullet$ The scalability experiments in Table 1 are conducted exclusively on synthetic 2D Gaussian point clouds, lacking validation on real-world large-scale datasets. More diverse and realistic datasets should be included to substantiate the scalability claims.

$\circ$ The paper could also benefit from improved table presentation. For example, although the experimental description in Section 5.2 and Table 1 are in close proximity, the main text does not contain an explicit citation and an explanation to Table 1.

---

> ### Author Rebuttal · Authors · 2026-03-29
>
> We thank the reviewer for the careful summary and for the positive assessment of both the geometry and the empirical behavior. The reviewer is right that the presentation around Table 1 can be improved; in the revision we will cite the table explicitly in the main text and explain the evaluation protocol more clearly.
>
> **On how the W_2 metric in Table 1 is computed.** We thank the reviewer for highlighting that this was under-specified in the current draft. We will make the definition explicit. Table 1 uses the 2-Wasserstein distance **between empirical distributions over coupling matrices**, with squared Frobenius norm as the ground cost between two couplings. Concretely, if $\widehat P=\frac1S\sum_{s=1}^S \delta_{\Gamma^{(s)}}$ and $\widehat Q=\frac1R\sum_{r=1}^R \delta_{\widetilde\Gamma^{(r)}}$ are empirical measures on the space of couplings, we compute
> $$W_2^2(\widehat P,\widehat Q) = \min_{\Lambda\in\Pi(a,b)} \sum_{s=1}^S\sum_{r=1}^R \Lambda_{sr}|\Gamma^{(s)}-\widetilde\Gamma^{(r)}|_F^2,$$ where $a=(1/S,\dots,1/S)$ and $b=(1/R,\dots,1/R)$. While Wasserstein distances are most commonly used to compare probability measures on data space, the definition is abstract and applies equally well to probability measures on any metric space; here the metric space is the set of coupling matrices equipped with the Frobenius norm. We will add this definition directly to the experimental section, since otherwise the table may be hard to interpret.
>
> So the 2-Wasserstein distances reported in Table 1 are computed between the method and the reference HMC baseline. In our experiments, $S=200$ for the evaluated SPVI/HMC empirical distributions, while $R$ is the for the evaluated SPVI/HMC empirical distributions, while $R=1000,1500,2500$ for $n=256,512,1024$, respectively. For MAP, the empirical measure is the Dirac mass at the MAP coupling.
>
> **On large-scale realism of the scalability experiment.**
> The reviewer is right that the current scalability benchmark is synthetic, and this was a deliberate design choice: our goal there was to isolate scaling in the coupling dimension and to compare directly against HMC/NUTS under controlled conditions. We will make this motivation clearer in the text. We also agree that real-world large-scale benchmarks would strengthen the paper further. Our present aim was to complement the synthetic scalability study with real applications in Office-31 and single-cell analysis, which probe different aspects of the method.
>
> **On absolute runtime.**
> The reviewer is right that, although SPVI is substantially faster than HMC, the absolute runtime is still nontrivial at the largest scales. The main computational bottleneck is the manifold volume correction term, which requires repeated Jacobian-vector products through truncated Sinkhorn. We will clarify this limitation in the revision and discuss possible accelerations, including implicit differentiation through the Sinkhorn fixed point, lower-dimensional parameterizations, and coordinate systems with simpler metric distortion. Such modifications could significantly improve efficiency, but would likely come at the price of moving away from the current chart’s particularly transparent geometric structure and its associated uniform finite-T convergence analysis as shown in Appendix A.

---

> > ### Author Rebuttal · Reviewer_CeoH · 2026-04-03
> >
> > Thank you for your response, I will keep my score.

---

### Official Review · Reviewer_SaDD · 2026-03-13

**Soundness:** 3
**Presentation:** 3
**Significance:** 2
**Originality:** 2
**Overall Recommendation:** 3
**Confidence:** 2

**Summary:**

The paper introduces Sinkhorn-parameterized Variational Inference for posterior estimation using polytopes. It demonstrates the scalability of proposed method with single-cell data and MNISTT dataset.

**Compliance With Llm Reviewing Policy:**

Affirmed.

**Key Questions For Authors:**

1. Is it possible tomprovide high dimensional experiments with celeba dataset 64x64?
2. Is it possible to show convergence time of the method and compare with other like LightSB in single-cell setup?
3. Could you explain what is the sense of proposititon 4.2?
4. Coulld you explain why do you  namely apply a bounded squashing map to ensure numerical stability?

**Limitations:**

no limitations

**Strengths And Weaknesses:**

strengths:

1. The paper uses Gibbs posteriors over couplings and propose promisable idea with pollytopes.
2. The paper is easy to follow

weaknesses:

1. The absence of high-dimensional datasets 64 x 64 like celeba of AFHQ. The MNIST dataset is not sufficient
2. Comparison with other methods in singlle-cell setup like LightSB and DSBM(Shroedinger bridges approach)
3. The absence of plot of convergence of the method.

---

> ### Author Rebuttal · Authors · 2026-03-29
>
> We thank the reviewer for the encouraging comments on the general idea and readability, and for the concrete suggestions.
>
> **On high-dimensional image datasets such as CelebA or AFHQ.**
>
> We agree that an additional experiment on a dataset such as CelebA or AFHQ could strengthen the empirical picture. At the same time, we would like to clarify what notion of “high-dimensionality” is most relevant for our method.
>
> Once the cost matrices $C^{(k)}$ have been constructed, the variational inference problem no longer depends directly on the ambient feature dimension $d$. SPVI operates on couplings $\Gamma \in \Pi(\mu,\nu)$, and the geometry of this inference problem is governed primarily by the support sizes $n,m$, through the intrinsic dimension $r=(n-1)(m-1)$ rather than by the raw feature dimension itself. In this sense, the principal notion of high dimensionality for our method is the dimension of the coupling space.
>
> That said, our current experiments already include a genuinely high-dimensional representation setting. In Office-31, the costs are built from four pretrained encoders, and the shared representation used by the classifier is the concatenation of their normalized embeddings, giving $d = 2048 + 768 + 768 + 512 = 4096$. Incidentally, this is already of the same order as a flattened $64\times64=4096$ image representation as in CelebA, and therefore substantially beyond the low-dimensional synthetic regime.
>
> **On comparison with LightSB / DSBM in the single-cell setting.**
>
> This is a useful suggestion, and it highlights an important distinction that we should make more clearly in the paper. Although Schrödinger bridge methods and our approach are closely related through entropic transport, they **do not** target the same inferential object. Schrödinger bridge methods such as LightSB or DSBM are designed to estimate a a full **interpolation in time**, or a bridge process, between two marginals, and whose **endpoint coupling** coincides with the corresponding entropic OT solution. Our method, by contrast, is designed to approximate **a distribution over endpoint couplings** induced by **uncertainty in the cost matrix**.
>
> Put differently, the randomness plays a different role in the two models. In Schrödinger bridge methods, it is the stochasticity of one entropic interpolation under fixed dynamics. In our setting, it reflects uncertainty over which endpoint alignment is plausible when the cost is itself not uniquely specified. For this reason, we do not view LightSB or DSBM as like-for-like posterior baselines, although they are certainly relevant comparison points for the single-cell application.
>
> **On convergence plots.** We agree. In revision we will add ELBO convergence and marginal residual versus iteration plots. We think this will make the optimization behavior much easier to understand.
>
> **On the meaning of Proposition 4.2.**
>
> Proposition 4.2 formalizes the fact that, after removing the natural gauge redundancy of the Sinkhorn representation, the interior of the transport polytope admits a genuine Euclidean coordinate system. The interior of the transport polytope is a constrained space, and the Sinkhorn representation has a built-in row/column redundancy in log-kernel coordinates. The proposition says that once this redundancy is removed, the remaining coordinates identify each strictly positive coupling uniquely and smoothly. In practical terms, this means that the model is not moving in an overparameterized ambient space: it is working in genuine intrinsic coordinates on the space of feasible couplings. That is exactly what allows us to use standard latent-variable tools, such as normalizing flows, while remaining inside the transport polytope and respecting the marginals by construction.
>
> **On the bounded squashing map.**
>
> We appreciate this question, because it made us realize that the current draft makes the squashing map appear more heuristic than it really is. The bounded squashing map is not needed for the existence of the exact Sinkhorn chart itself, but it plays three useful roles in the practical method. First, it is important numerically since the map passes through $K=\textrm{exp}(B)$, the unbounded logits can lead to severe overflow or underflow, and can also make Sinkhorn scaling and the Jacobian Gram term poorly conditioned. Secondly, in the global chart, moving to large $|z|$ corresponds to approaching the sparse boundary of the transport polytope. Bounding $z$ keeps the variational family in a compact, well-behaved subset of the interior. Third, our finite-$T$ analysis of truncated Sinkhorn in Appendix A uses compactness of the latent domain to obtain uniform kernel bounds, which in turn yield the uniform contraction rate.

---

> > ### Author Rebuttal · Reviewer_SaDD · 2026-04-01
> >
> > Thanks a lot for your answers. Nonetheless, I am totaly sure that additionaly experiment with Celeba or FFHQ. should be done, excuse me, but form point if my view, it is necessary to detect the force of your method

---

> > > ### Author Response · Authors · 2026-04-02
> > >
> > > Thank you again for your follow-up. We appreciate your point that a face benchmark would make the practical strength of the method more concrete. We took this suggestion seriously and ran two additional experiments with the aligned CelebA images resized to $64\times64$ and a setup with $n=m=128$ per OT problem with $\mu_i=1/n$ and $\nu_j=1/m$, hence $\Gamma\in\mathbb{R}^{128\times128}$. The source set contains $128$ faces satisfying a fixed set of invariant attributes, while the target set is a mixture of two semantic modes, with $64$ targets from each mode. The source and target identities are kept disjoint using the annotations provided by CelebA. Similar to the Office-31 setup, for the cost construction we use six frozen encoders: ArcFace[1], FaceNet[2], DINOv2, CLIP, DeiT, FAN[3]. For the FAN, we use the penultimate layer representation. If one concatenates the six normalized embeddings into a single shared representation, the resulting feature dimension is $512+128+768+512+768+512=3200$. However for the posterior itself, the important objects are not the raw embeddings but the six pairwise cost matrices $C^{(k)}\in\mathbb R^{128\times128}$ they induce. We normalize each cost matrix by its median value so no single view dominates by scale alone.
> > >
> > > The first experiment has 3 tasks intended to test the core claim of the paper that the posterior captures semantic ambiguity and avoids collapsing to one target regime. The first task maps young smiling women with brown hair to a target mixture of blond-hair and black-hair women. The second maps young clean-shaven men to a target mixture of mustache and goatee styles. The third maps young women with black hair and no hat to a target mixture of straight-hair and wavy-hair styles, again keeping black hair and no hat fixed. We report only the quantitative metrics here and will include the corresponding figures in the revised draft. We use $S=50,000$ posterior samples to compute the Fréchet Inception Distance using `pytorch-fid` after resizing wrt **all** valid target images, not just for the $128$ images used for the OT target and the invariant attribute accuracy is computed using FAN-based attribute predictions. Values are averaged over 5 random seeds. We use 8 layers with $S=8, M=8, L=20$. The SPVI runtime <10 mins, BayesOT <30 mins for each task, consistent with our scalability experiment.
> > >
> > > |Method|Hair-color FID|Hair-color InvAcc|Facial-hair FID|Facial-hair InvAcc|Hair-style FID|Hair-style InvAcc|
> > > |-|-:|-:|-:|-:|-:|-:|
> > > |SPVI|9.4±0.7|0.94±0.05|8.3±0.6|0.93±0.04|8.8±0.6|0.96±0.07|
> > > |BayesOT|6.1±0.4|0.96±0.02|6.2±0.5|0.94±0.05|6.8±0.7|0.96±0.03|
> > >
> > > For the second experiment, we follow the same joint transport-and-classifier learning protocol as in Office-31. We define three subdomains within CelebA, which we refer to as Clean, Cosmetic, and Wavy. The Clean domain consists of faces without hats, eyeglasses, heavy makeup, or lipstick. The Cosmetic domain consists of faces without hats or eyeglasses but with visible cosmetic styling, namely heavy makeup and/or lipstick. The Wavy domain consists of faces without hats, eyeglasses, heavy makeup, or lipstick, but with wavy hair. This gives us a controlled family of appearance shifts while keeping the underlying facial semantics comparable. For prediction, we use a disjoint set of six CelebA attributes- Smiling, Male, Blond Hair, Black Hair, Mustache, and Young, so that the labels being evaluated are not the same attributes used to define the source-target split (otherwise the task is circular). We use a one-vs-all squared hinge loss on signed labels applied independently to each of the selected binary attributes, yielding a multi-label attribute classifier. The prediction label is computed by averaging over $q_\phi(\Gamma)$.
> > >
> > > |Domains|Base|SPVI|Runtime(s)|
> > > |-|-:|-:|-:|
> > > |Clean→Cosmetic|81.22|89.38|4021|
> > > |Cosmetic→Clean|79.27|90.48|4176|
> > > |Clean→Wavy|80.24|85.87|4380|
> > > |Wavy→Clean|81.10|85.56|4475|
> > > |Cosmetic→Wavy|76.24|81.23|4984|
> > > |Wavy→Cosmetic|75.90|79.14|4865|
> > >
> > > So in every transfer direction, the posterior transport model improves substantially over the base classifier. Since this is a joint optimization, each domain transfer takes around an hour on CPU (excluding the one-time extraction of encoder features from the raw images). Given the rebuttal-time constraints, we avoid multiple seeds and tuning to choose the optimal $\alpha, \lambda_{\Omega}, \lambda_s$ and use the hyperparams from our initial Office-31 experiment. In the revision we will include the fully tuned and averaged version, together with qualitative figures and encoder repository links for reproducibility.
> > >
> > > Thank you again for pushing us on this point, we believe this addition would strengthen the claims of our paper and might make our work more relatable to the computer vision community as well.
> > >
> > > References:
> > >
> > > [1] http://dx.doi.org/10.1109/TPAMI.2021.3087709
> > >
> > > [2] http://dx.doi.org/10.1109/CVPR.2015.7298682
> > >
> > > [3] https://doi.org/10.24963/ijcai.2018/102

---

### Official Review · Reviewer_BKxQ · 2026-03-13

**Soundness:** 3
**Presentation:** 3
**Significance:** 3
**Originality:** 3
**Overall Recommendation:** 4
**Confidence:** 3

**Summary:**

This paper studies inference over entropic transport plans. The main idea is to use the Sinkhorn map as a  parameterization of the transport map. Empirically, the paper is quite strong, with experiments on  modeling, scalability, domain adaptation, and single-cell dynamics.

**Compliance With Llm Reviewing Policy:**

Affirmed.

**Final Justification:**

All questions are well addressed, and I would be happy to maintain my score (4).

**Key Questions For Authors:**

1. What do the authors view as the main theoretical novelty of the paper?
2. Do the authors have evidence that the method remains effective in genuinely high-dimensional feature spaces, not only for larger transport problems?
3. Why should this approach be particularly suitable for domain adaptation? Can the authors provide a clearer theoretical foundation for the domain adaptation application, especially regarding the role of uncertainty over transport plans?

**Limitations:**

Yes

**Strengths And Weaknesses:**

A main strength is that the paper addresses a meaningful problem. The empirical results are also solid and diverse, and the scalability comparison is particularly useful. My main concern is that the theoretical contribution feels more technical than conceptually new. The results seem to justify the proposed parameterization and approximation scheme, but it is less clear what the theoretical novelty is beyond making the framework well-defined and tractable.

---

> ### Author Rebuttal · Authors · 2026-03-29
>
> We thank the reviewer for the careful reading and for capturing both the empirical strengths and the current gap in how we articulated the theory.
>
> **On the main theoretical novelty.**
>
> We appreciate this question, because in the current draft the theory is probably presented in a more technical way than necessary. The point we regard as most conceptually important is actually quite simple. The space of admissible couplings is not an unconstrained Euclidean space; it is the interior of the transport polytope, so any variational method has to respect the marginal constraints exactly. What our construction shows is that, after removing the natural row/column redundancy of the Sinkhorn parameterization, one obtains a genuine global coordinate system for that interior. In other words, the method does not merely produce feasible couplings, it turns the interior of the transport polytope into something one can parameterize and optimize over as a smooth latent space of the correct intrinsic dimension.
>
> This has an important consequence for variational inference. Once the space of couplings is viewed as a manifold in its own right, the relevant density is not the ambient Euclidean density but the intrinsic density on that manifold. The Gram determinant term in the ELBO is precisely the correction associated with that change of geometry. In our view, this is the main conceptual contribution of the theory: not simply a parameterization that respects the constraints, but a principled way to place and optimize distributions directly on the space of transport plans.
>
> **On genuinely high-dimensional feature spaces.**
>
> The reviewer is correct, we emphasize on the scaling in transport size than on scaling in ambient feature dimension. It is important to distinguish these two notions of dimension. More precisely, once the cost matrices $C^{(k)}$ have been constructed, the ambient feature dimension $d$ no longer enters the variational inference problem directly. SPVI performs inference over couplings $\Gamma \in \Pi(\mu,\nu)$, whose geometry is governed by the support sizes $n,m$ through the intrinsic dimension $r=(n-1)(m-1)$, not by the original feature dimension. In this sense, the principal notion of “high-dimensionality” for our method is the dimension of the coupling space rather than the dimension of the raw features.
>
> The most relevant evidence we currently provide is the Office-31 experiment, where the costs are built from four modern pretrained encoders and a shared classifier on the concatenated normalized embeddings, which has an ambient dimension $d=2048+768+768+512=4096$. So the method is already being used with genuinely high-dimensional learned representations, not only low-dimensional synthetic features. However, we agree that this is not the same as a controlled experiment isolating growth in $d$ while fixing $(n,m)$. We will revise the claims to reflect this distinction clearly.
>
> **On domain adaptation and why uncertainty over transport plans matters there.**
>
> This is an important point, and we appreciate this point; it helped us see that the current text did not explain the scientific motivation sufficiently clearly.
>
> In domain adaptation, transport is used as a latent alignment mechanism between source and target samples. A deterministic OT solver returns a single alignment for one chosen cost matrix. When several cost matrices are scientifically plausible, for example because different feature representations induce different cross-domain distances, or because some source points admit several comparably good target matches, the selected coupling can change discontinuously across these plausible geometries. Our posterior over couplings is designed to represent exactly this variability in the alignment, rather than forcing commitment to one transport plan. In the Office-31 setting, different frozen encoders provide distinct views of cross-domain geometry, so the uncertainty is a direct reflection of ambiguity in representation space.
>
> The role of uncertainty in the classifier update is also structural. If $\hat P(\Gamma)=\Gamma^\top P^s$ denotes the soft target labels induced by a coupling, then the classifier is trained against the posterior predictive target labels obtained by averaging over $q_\phi(\Gamma)$, rather than labels induced by a single brittle alignment. So instead of committing to one single transport plan, we average over plausible alignments, which is especially useful when class overlap or representations shift makes several couplings credible. We will expand this discussion in the domain adaptation section.

---

> > ### Author Rebuttal · Reviewer_BKxQ · 2026-04-03
> >
> > Thank the authors for the response. I read the authors' rebuttal, and all questions are well addressed, and I would be happy to maintain my score.

---

> > > ### Author Response · Authors · 2026-04-04
> > >
> > > Dear Reviewer,
> > > thank you for your answer and your engagement.
> > > We appreciate you confirming that all your concerns are now fully resolved. Since the issues raised in the initial review have been addressed, we kindly ask if you would consider increasing the score. If there are any remaining points that still stand in the way of a higher rating, please let us know so we can address them immediately.

---

### Decision · Program_Chairs · 2026-04-30

**Decision:**

Accept (regular)

**Comment:**

The authors propose Sinkhorn-parameterized Variational Inference (SPVI) which is a novel approach for posterior inference over optimal transport plans under uncertain ground costs. The main idea is to leverage a gauge-fixed Sinkhorn map as a differentiable reparameterization of the transport polytope, which enables variational inference with expressive models (e.g., via normalizing flows) while enforcing marginal constraints.

We think the proposed framework is conceptually interesting with encouraging empirical results (e.g., its speedup over Hamiltonian Monte Carlo (HMC)). However, the reviewers also raised concerns on its computational cost (i.e., runtime remains relatively high at large scale despite its speedup over HMC), its empirical validation (e.g., rely heavily on synthetic or small-scale datasets), and presentation clarity. Overall, we think while there are still concerns on clarity and empirical coverage, its contributions are solid and interesting for the community on uncertainty-aware optimal transport. Therefore, we recommend acceptance.